

# The impact of soil development, rainfall intensity and vegetation complexity on subsurface flow paths along a glacial chronosequence of 10 millennia

Anne Hartmann[1], Markus Weiler[2], Konrad Greinwald[3], and Theresa Blume[1]

[1]Section Hydrology, GFZ German Research Centre for Geosciences, Potsdam, Germany
[2]Faculty of Environment and Natural Resources, Chair of Hydrology, University of Freiburg, Freiburg, Germany
[3]Department of Geobotany, Faculty of Biology, University of Freiburg, Freiburg, Germany

**Correspondence:** Anne Hartmann (aha@gfz-potsdam.de)

**Abstract.**

Hydrologic processes play an important role in the hydro-pedo-geomorphological feedback cycle of landscape evolution. Soil properties and subsurface flow paths change over time, but due to lack of observations important hydrologic processes such as water flow paths are often not properly considered in soil and landscape evolution studies. We investigated the evolution
of subsurface flow paths during landscape development in the calcareous glacier forefield at the Griessfirn in the Swiss Alps. While the main focus was on flow path evolution and the formation of preferential flow paths with soil development, we also looked at the impact of irrigation intensity and vegetation complexity (in what way does the vegetation complexity defined by degree of vegetation cover and functional diversity at each age class relate to subsurface structures and flow path initialization?). We chose four glacial moraines of different ages (110, 160, 4 900, and 13 500 years) and conducted dye tracer experiments
with Brilliant Blue (4 g $l^{-1}$) on three plots at each moraine. The three plots at each age class differed by their degree of vegetation complexity (low, medium, and high) and each was further divided into three equal subplots where dyed water was applied with three different irrigation intensities (20, 40, and 60 mm $h^{-1}$) and an irrigation amount of 40 mm. Dye pattern characteristics in form of volume density and surface area density were derived by digital image analysis and compared via statistical analysis.Volume density was used to classify the observed dye patterns into specific flow type categories. The effect
of soil formation and thus changing soil characteristics on flow types were investigated by the analysis of structural and textural parameters in form of grain size distribution, porosity, bulk density, and loss on ignition. A change in flow types with increasing moraine age was observed from a rather homogeneous matrix flow to heterogeneous matrix and finger flow. Along the soil chronosequence, a reduction in particle sizes and an ongoing vegetation development resulted in an accumulation of organic matter in the topsoil and an increase in water storage capacity (decrease in bulk density and increase in porosity).
Differences in irrigation intensity only had an effect on flow types at the oldest moraine, where the frequency of finger flow decreased with increasing irrigation intensity. A relation between vegetation complexity and flow types was only observed at the older moraines, which had a dense vegetation cover. With increasing vegetation complexity the proportion of preferential flow paths in form of finger flow also increased.



## 1 Introduction

Soil development and its interaction with hydrological, geomorphological and ecological processes plays a crucial and complex role within landscape evolution. Changes in soil structure and texture due to pedogenesis (including vegetation development) leads to changes in soil hydraulic behavior and thus to changes in water flow paths. This also includes the development of preferential flow paths. The change in hydrologic processes and water fluxes in turn affect pedogenic processes (e.g. weathering, solute transport) and geomorphic processes (e.g. water erosion) (van der Meij et al., 2018). Changes in hydrologic flow paths impact the water budget, overland flow generation, sediment transport, flood generation and water availability. Especially preferential flow has a strong impact on water storage (Rye and Smettem, 2017), plant water availability, and the transport of nutrients and contaminants (Jarvis (2007), Jin and Brantley (2011), Bundt et al. (2000)) and thus on groundwater quality (Alaoui, 2015). Preferential flow is, however, not only controlled by soil properties (Koestel and Jorda, 2014) but also by rainfall characteristics such as amount (Bachmair et al., 2009) and intensity (Wiekenkamp et al., 2016), with preferential flow occurring more often at higher rainfall intensities (Gjettermann et al. (1997), Lin and Zhou (2008), Liu and Lin (2015), Demand et al. (2019)). However, according to van der Meij et al. (2018) there is a lack in proper handling of hydrologic processes and their role within the feedback cycle of the hydro-pedo-geomorphological system when it comes to soil and landscape evolution modeling. Especially preferential flow is often not considered in soil evolution modeling studies. Among other difficulties this is related to limited data and observations of how flow paths and soil properties change over time. This could be remedied with systematic chronosequence studies.

Chronosequence studies, where time is substituted by space, are frequently used to study soil development (Crocker and Dickson (1957), Douglass and Bockheim (2006), He and Tang (2008), Egli et al. (2010), Dümig et al. (2011), Vilmundardóttir et al. (2014)). Most of the studies focused mainly on changes of soil biological, chemical and physical properties during soil development by studying sites at different stages (Crocker and Major (1955), D'Amico et al. (2014), Hudek et al. (2017), Musso et al. (2019)) and estimating development rates. The impact of soil properties development on the soil's hydraulic behavior and especially the impact on water transport and preferential flow paths was only investiagted by a few studies (Lohse and Dietrich (2005), Yoshida and Troch (2016), Hartmann et al. (2020a)). The overall observed changes in soil texture and structure are not only linked to physical and chemical processes, but also to vegetation development (Morales et al. (2010), Hudek et al. (2017)). Vegetation characteristics in form of type (Bachmair et al., 2009) and complexity (Ossola et al., 2015) influence hydrologic processes.

Dye tracer experiments have been applied successfully to study preferential infiltration in soils under different land covers (Blume et al. (2008), Bachmair et al. (2009), Bogner et al. (2014)). In a previous study we had investigated how water flow paths evolve along with soil formation over 10 millennia on siliceous parent material by using Brilliant Blue dye tracer irrigation experiments (Hartmann et al., 2020a). The observed flow types changed from a rather homogeneous rapid matrix flow in coarse material at the youngest moraine to a mainly finger flow at the medium-age moraines. At the oldest moraine, only macropore



flow via root channels was observed in deeper parts of the soil, in combination with a very high water storage capacity of the organic top layer. In general, we found an increase in water storage with increasing moraine age. Preferential flow was, however, not only caused by macropores, but especially for the medium age moraine seemed to be mainly initiated by soil surface characteristics (vegetation patches and micro-topography).

The here presented follow-up study differs from the previous study in that it focuses on i) evolution processes on calcareous parent material, ii) the impact of different irrigation intensities (i.e. does the impact of rainfall intensity on subsurface flow paths change across the millennia?) and iii) the influence of vegetation complexity (in what way does the vegetation complexity at each age class relate to subsurface structures and flow path initialization?). We assume that a strong link between vegetation, soil properties, and subsurface flow paths exists. Vegetation complexity combines both, surface characteristics (vegetation

coverage) and functional aspects of the vegetation (Greinwald et al., 2021a) in one measure. Hydrologists often link indicators such as vegetation coverage and land use to hydrological functioning as they are likely to affect water infiltration patterns and soil water transport (Bachmair et al., 2009). Vegetation type and cover influence the soil water budget and as such the role of water within the weathering processes. Chemical and physical root activities have an impact on the physical and chemical weathering processes and thus on the physical soil properties, but also on the channeling of soil water (Amundson et al., 2007).

Vegetation composition also had a strong influence on weathering intensities in the tropics (Ivory et al., 2014). In our study we move beyond simple indeces such as % coverage and land use type and take a more detailed view of habitat composition and underground plant traits, e.g. root type and root growth form, (lumped together in the vegetation complexity index) will be helpful in interpreting differences in subsurface characteristics and flow paths.

To investigate possible relationships of age, irrigation intensity and vegetation complexity with the dye patterns we applied the

bootstrapped LOESS regression (BLR) approach by Keith et al. (2016). The approach was originally proposed for the analysis of differences between soil property-profiles and has not been previously applied for the analysis of Brilliant Blue experiments. This approach enables a depth-dependent analysis of differences among the dye coverage profiles and surface area density profiles.

Our research, which focuses on the evolution of hydrological flow paths across a chronosequence in the Swiss Alps, is part

of a larger interdiscplinary chronosequence study covering aspects of geomorphology (Musso et al. (2019), Musso et al. (2020)), geobotany (Greinwald et al. (2021a), Greinwald et al. (2021b)), and surface (Maier et al., 2020) and subsurface hydrology (Hartmann et al. (2020a), Hartmann et al. (2020b)). This interdisciplinary study opens up a broader view of the various processes either driving the feedback loop of landscape evolution or affected by it.

## 2   Material and methods

### 2.1   Study site

The study area at the Griessfirn forefield is located above the treeline between 2030-2200 m a.s.l. in the Central Swiss Alps (appr. 46° 85'N, 8° 82'E). The geology is dominated by schists, marl, and quartzites (Musso et al., 2019), but also includes limestone (Frey, 1965). The closest official weather station located at a similar elevation (2106 m a.s.l.) is 48 km away at





Mount Pilatus (46° 98'N, 8° 25'E). The recorded annual mean temperature is 1.8 °C and the annual precipitation is 1752 mm (1981-2010) (Swiss Confederation, 2020).

The four selected moraines were dated by Musso et al. (2019) based on historical maps and additional radiocarbon dating. The youngest moraine is located at 2200 m a.s.l. and was dated to an age of 110 years. The three other moraines are 160, 4 900,

5  and 13 500 years old and are located at an elevation of 2030 m a.s.l. The two oldest moraines are densely covered with grass (e.g. *Festuca violacea*), dwarf shrubs (e.g. *Rhododendrum hirsutum, Vaccinium myrtillus, Vaccinium vitis-idaea*) and sedge. The coverage of the two younger moraines is sparse with carpet forming dwarf shrubs (e.g. *Dryas octopetala, Salix retusa*) stabilizing the slopes by their dense root stocks and facilitating the establishment of other plant species (such as *Anthyllis vulneraria, Saxifraga oppositifolia, Silene acaulis*) at the 160-year-old moraine and patches of pioneer plants (e.g. *Saxifraga*

10  *aizoides*) and mosses (e.g *Tortella densa, Distichium capillaceum*) at the 110-year-old moraine.

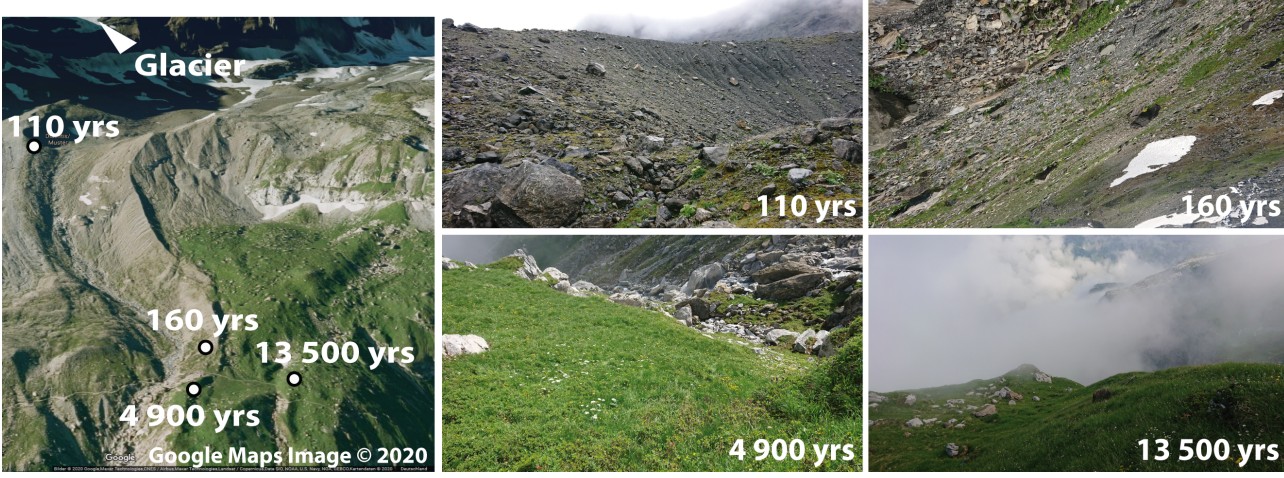

**Figure 1.** Location of the four selected moraines in the Griessfirn forefield (left, photo by © Google (2020)) and the surface cover of each age class (right).

## 2.2  Plot selection and vegetation complexity

Three plots (1.0 x 1.5 m) were selected on each moraine for the brilliant blue tracer experiments. At each plot the structural vegetation complexity measure (Greinwald et al., 2021a) was estimated. This measure was developed in the larger context of the interdisciplinary chronosequence study (e.g. Musso et al. (2019), Maier et al. (2020), Greinwald et al. (2021b)) and is

15  defined as a combination of vegetation cover and a measure of functional diversity. We chose the functional divergence as an index of functional diversity, which describes the variance of species traits in the community (Greinwald et al., 2021b). This index describes the functional similarity of communities in the multivariate niche space, and it has the advantage of not being correlated to vegetation cover. At each plot all vascular plant species and vegetation cover were recorded. With this data on site-community assemblage and data on the plant functional traits, we calculated the functional diversity. To do so,



we used the following eight functional traits characterizing species along the main axes of plant performance (Garnier et al., 2016): specific leaf area, nitrogen content, leaf dry matter content, Raunkiær's life form, seed mass, clonal growth organ, root type, and stem growth form. Specific leaf area, leaf nitrogen content, and leaf dry matter content were measured by using the guidelines of the trait handbook by Pérez-Harguindeguy et al. (2013). Seed mass was extracted from the Seed Information

Database (Royal Botanic Gardens Kew, 2019). Raunkiær's life form, clonal growth organ, and stem growth form were extracted from the European trait database LEDA (Kleyer et al., 2008). The determination of root types was based on field observations and literature surveys (Kutschera et al. (1991), Kutschera and Lichtenegger (1992), Kutschera and Lichtenegger (2013)); we applied the classification scheme of Marden et al. (2005). We used the data on functional divergence as well as the cover data to conduct a principal component analysis and defined the scores of the first axis as an index of functional complexity.

Additionally, root length density (RLD), root density (RD), and specific root length (SRL) of the fine roots (diameter < 1 mm), as well as above ground biomass (BM) were measured as described in Greinwald et al. (2021a) at each experiment plot.

To investigate the soil properties, three plots for soil sampling were selected on each moraine along a within-moraine vegetation complexity gradient. At the 110-year-old moraine, the soil samples were taken from within the tracer experiment plots. At the other three moraines, soil samples were taken at two locations within three 4 x 6 m plots per moraine, which were additionally

part of larger scaled irrigation experiments. The vegetation complexity of these plots was determined using the same method, only that it was determined on a large scale covering the entire moraine. Therefore, at each moraine the vegetation was mapped by differentiating between vegetation units, which we defined as stocks of vegetation classified by the dominating species. We further characterized each unit by vegetation surveys (10 x 1 m$^2$) recording all vascular plant species and cover. The vegetation surveys were randomly distributed within the mapped vegetation units. We sorted the vegetation surveys in ascending order

of the index of functional complexity and selected the vegetation surveys with highest, intermediate and lowest vegetation complexity as the plot locations. The exact plot dimensions of 4 x 6 m were then determined by up-scaling around the center of the selected locations. At the soil sampling plots, the scale of the determined vegetation complexity therefore exceeds the scale of the soil sampling locations (except at the 110 year-old moraine), while at the tracer experiment plots the scale of the experiment is identical to the scale of the vegetation complexity.

**2.3  Field experiments**

The dye tracer experiments were conducted between August and mid September 2019 with a 4 g l$^{-1}$ Brilliant Blue FCF solution. Each of the three 1.5 m x 1.0 m experiment plots per moraine was further divided into three equal subplots of 0.5 m x 1.0 m for individual irrigation with 40 mm of dyed water (Figure 2). To reduce interception, large vegetation in form of shrubs, bushes, and tall grass was cut to a height of a few centimeters before irrigation. The three subplots were irrigated with the same

amount (40 mm) but different intensities (20, 40, and 60 mm h$^{-1}$). The irrigation intensities represent extreme events with return periods of 2.8, 60, and 100 years (Fukutome et al., 2017), respectively.





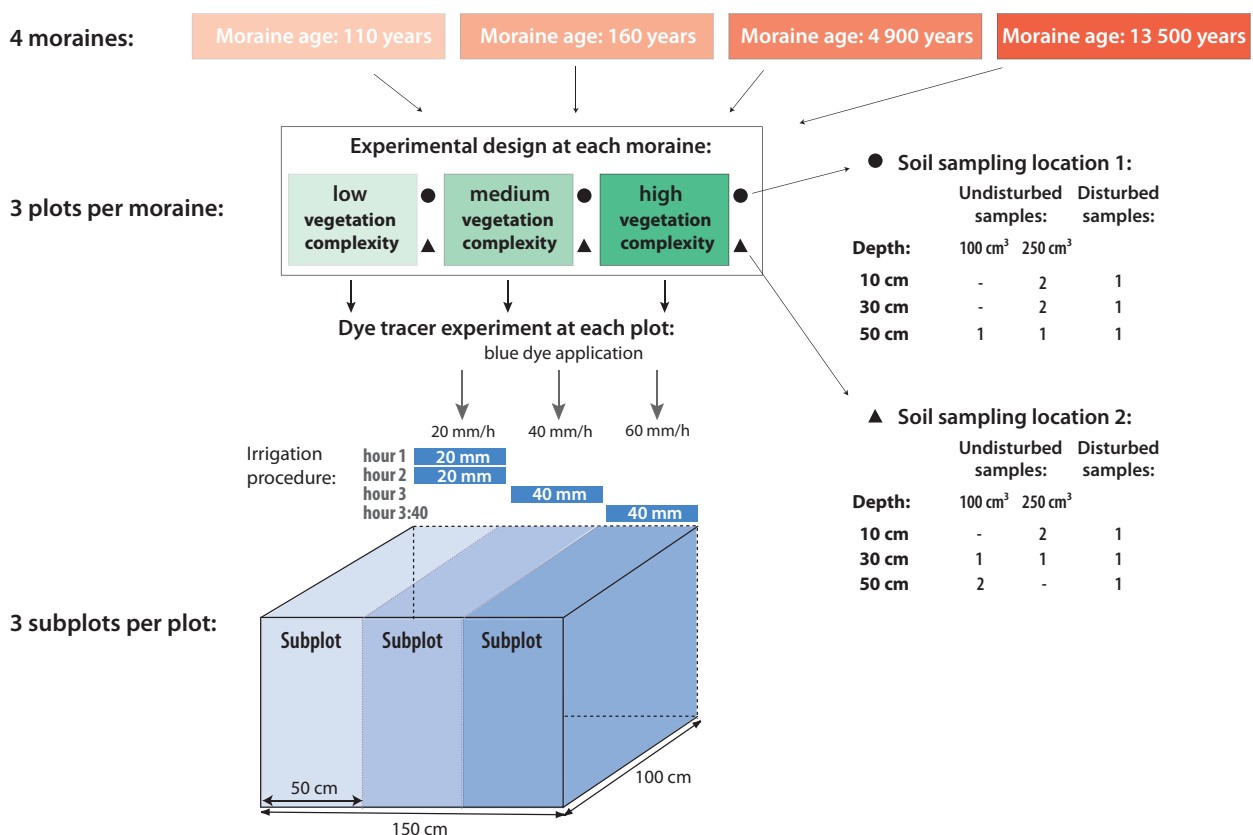

**Figure 2.** Illustration of the experimental design and soil sampling scheme at each moraine (adapted from Hartmann et al. (2020a)).

Each subplot was irrigated individually, while the others were covered by a tarpaulin. A hand-operated sprayer and a battery-powered pump were used for tracer application. Since the irrigation system only provided a flow rate of $1\,l\,\text{min}^{-1}$ the different irrigation intensities were achieved by alternating intervals of irrigation and breaks. The first subplot was irrigated for 2 hours in a sequence of 1 minute irrigation and 5 minutes break to irrigate the subplot with 40 mm at an intensity of $20\,\text{mm}\,\text{h}^{-1}$.

5 The intensity of $40\,\text{mm}\,\text{h}^{-1}$ at the second subplot was achieved by a sequence of 1 minute irrigation and 2 minutes break for 60 minutes. The last plot was irrigated for 40 minutes in a sequence of 1 minute irrigation and 1 minute break to achieve an intensity of $60\,\text{mm}\,\text{h}^{-1}$. Figure 2 shows the experimental design at each moraine and illustrates the irrigation procedure. After the experiment the whole plot was covered until excavation on the following day.

The plots were excavated in five vertical profiles in approximately 10 cm segments starting from the lower edge of the irrigated

10 plot. Pickaxes, spades, and hand shovels were used to excavate the profiles. The profiles were cleaned carefully and roots were cut off. Rocks and stones were not removed, but cleaned from soil. The soil profiles of the subplots were photographed with a Panasonic Lumix DMC-FZ18 camera and a resolution of 2248 pixels x 3264 pixels. To avoid direct sunlight and provide a





uniform light distribution a large umbrella was used for shading. A Kodak-gray-scale and a wooden frame were included in the photographs for a later geometric correction and color adjustment.

## 2.4 Image analysis and flow type classification

We used the same image analysis and flow type classification as described in Hartmann et al. (2020a). Tricolor images were
generated based on the photographs by using the image analysis procedure by Weiler (2001). A geometric correction, a background subtraction and color adjustment was carried out to correct differences in image illumination and changes in the spectral composition of the daylight. Further details on the method can be found in Weiler and Flühler (2004). The delineation of rocks and plants was done manually. In the resulting tricolor image the horizontal and vertical length of a pixel correspond to 1 mm. A further correction of the tricolor images using the photographs was necessary, since due to poor lighting conditions or a
heterogeneous background color distribution in the soil caused by e.g. material transitions, small stones or organic matter, the image analysis software was not able to recognize all large dye stains as coherent objects. Thus, the software detected interruptions within blue stains that did not correspond to the field observations and would have been identified as a large number of individual flow paths during the following analysis. The manual correction of the images using the photographs eliminated these interruptions (Hartmann et al., 2020a).

The volume density corresponds to the dye coverage and was calculated for each of the five profiles per subplot as the fraction of stained pixels at each depth. The volume density profiles were then averaged of all excavated profiles per subplot. The surface area density was calculated by using the intercept density, which describes the amount of interfaces between stained and unstained pixels divided by the horizontal width of the soil profile. The profiles of the surface area density were also averaged over all five photographed profiles per subplot. The combination of both profile parameters provides information whether the
stained area is the sum of many small fragments or a few large ones.

The resulting dye patterns were then classified into flow type categories according to the approach proposed by Weiler (2001). This is based on the proportions of three selected stained path width classes (stained path width <20 mm, 20 mm-200 mm, >200 mm) relative to the volume density. The stained path width is equal to the horizontal extent of a stained flow path. This classification method distinguishes between five flow types: (1) macropore flow with low interaction, (2) mixed macropore
flow (low and high interaction), (3) macropore flow with high interaction, (4) heterogeneous matrix flow/finger flow, and (5) homogeneous matrix flow. Dye patterns, which could not be classified as one of these flow types were categorized as undefined. We used a modified version of this classification by Hartmann et al. (2020a) which was more suitable for stony alpine soils. The modified classification avoids that homogeneously blue stained areas are classified as smaller stained path widths due to the interruption by rocks. In this case the flow type is assigned to a new flow type class called 'homogeneous matrix flow
between rocks' which basically can also be assigned to the flow type class 'homogeneous flow' of the original classification. The modified classification also avoids a clear differentiation between 'macropore flow with high interaction' and 'finger flow'. This is based on the observation that finger-like flow paths with smaller widths were frequently present in these alpine soils which otherwise would have been misclassified as 'macropore flow'.





## 2.5 Soil sampling and laboratory analysis

Soil samples were taken during August and September of 2019. The samples at the 110-year-old moraine were taken directly within the non-excavated part of the experimental plots. At the other moraines, samples were taken nearby at larger plots (4 x 6 m). At each plot, soil samples were taken at two locations within the plot. The two sampling locations were less than 1 m

apart at the 110-year-old moraine and ≈ 2 m apart at the other moraines. The sampling scheme is also visualized in Figure 2. Two disturbed soil samples per depth were taken in 10, 30, and 50 cm depth at each plot. The analysis of the total of 72 samples was carried out in the laboratory between November 2019 and January 2020. We used the laboratory analysis methods as described in Hartmann et al. (2020a) and Hartmann et al. (2020b). A combination of dry sieving (particles > 0.063 mm) and sedimentation analysis (particles < 0.063 mm) with the hydrometer method (Casagrande, 1934) was used. Particles between

2 mm and 0.063 mm were classified as sand, between 0.063 mm and 0.002 mm as silt and particles smaller than 0.002 mm as clay. Organic matter removal was due to lab limitations only possible by floating off the lighter fractions prior to particle size analysis and by combustion of the samples prior to dry sieving. Furthermore, the use of acid-solutions for organic matter removal would also have carried the risk of a simultaneous removal of calcite from the calcareous soil samples. A maximum temperature of 550 °C was used for combustion to avoid effects on the calcite fraction (Hoogsteen et al., 2018).

At each plot four 250 cm$^3$ undisturbed soil samples were taken at a depth of 10 cm. In 30 cm depth three 250 cm$^3$ and one 100 cm$^3$ and in 50 cm depth one 250 cm$^3$ and three 100 cm$^3$ undisturbed soil samples were taken. This sampling scheme provides 4 undisturbed soil samples per depth and thus 12 samples per study plot to examine bulk density, porosity and loss on ignition (LOI). The porosity was determined by using the water saturation method. For this method, sample weights were recorded at saturation and after drying at 105 °C. For saturation, the samples were placed in a small basin. The water level in

the basin was increased step-wise by 1 cm d$^{-1}$ to minimize air entrapment. When the water level reached the top of the soil sample and the sample was fully saturated, the bottom of the sample was sealed, and the weight at saturation was measured. Bulk density was determined by relating the dry mass after drying at 105 °C to the sample volume. The LOI was estimated by drying sub-samples (4-8 g) of all (undisturbed) soil samples for at least 24 hours at 105 °C and 550 °C. The weight loss after drying at 550 °C relative to the weight after drying at 105 °C is specified as LOI and is a measure for the organic matter

content.

## 2.6 Statistical analysis

A bootstrapped LOESS regression (BLR) approach (Keith et al., 2016) was used to test differences in the observed averaged dye coverage (volume density profiles) and surface area profiles with regard to moraine age, irrigation intensity, and vegetation complexity. The BLR approach is a combination of bootstrapped data resampling with local least-squares-based polynomial

smoothing (LOESS) regression and was proposed by Keith et al. (2016) for the comparison of any soil property profiles from data sets of two different characteristics (e.g. age). For the pair-wise test the two data sets containing several profile observations of the soil variable of interest are combined and resampled n=1000 times by bootstrap with replacement. Each resampled data set is modelled using LOESS regression. Out of the 1000 LOESS regressions the 95 % confidence intervals are calculated and





compared with the LOESS regression of the original data sets. When the modelled LOESS regression of the original data set lies outside of the confidence interval, the null hypothesis, that there is no difference between the two original data sets, is rejected.

To test for significant differences of the soil physical characteristics among the age classes and the complexity levels the Tukey's
HSD (Honestly Significant Difference) test was used. The significance level was set to $p < 0.05$. The Turkey test compares all possible pairs of mean values and determines which mean values differ from the rest. Differences between the observations are considered significant if the difference in the mean values is higher than the expected standard error. All data analyses were carried out using R (R Core Team, 2017).

## 3   Results

### 3.1   Vegetation complexity

The obtained complexity values range between -0.79 (indicating high complexity) and 0.78 (indicating a low complexity). For a better visualization and comparison the values were inversely normalized to a range from 0 to 1.0, that values close to 0 correspond to a low complexity and a values close to 1 to a high complexity. Figure 3 shows the normalized values of the vegetation complexity for the three soil sampling plots (a) and the three tracer experiment plots (b) per moraine. The
values can only be compared with each other within the respective plot type, since they were obtained by separate principal component analyses. At the soil sampling plots, the levels of complexity differ only very slightly at the oldest moraine. At the other age classes, the differences between the medium and high complexity are also very small. Therefore, a valid evaluation of the results with regard to the degree of vegetation complexity is primarily restricted to the three youngest moraines and a comparison of the results at the low complexity plot with the results at the other two plots. At the tracer experiment plots, the
differences between the complexity levels are clearly pronounced at the 13 500- and 160-year-old moraine. At the 110-year-old moraine only the low complexity level is clearly different from the other two levels and at the 4 900-year old moraine only the high complexity differs clearly from the low and medium complexity level.





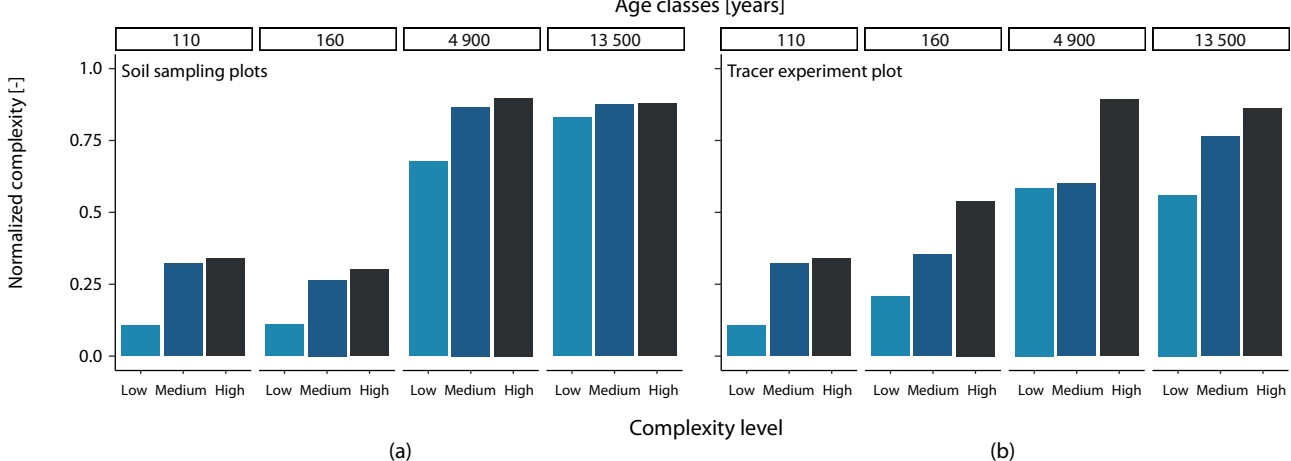

**Figure 3.** Normalized vegetation complexity values of the three complexity classes at the soil sampling plots (a) and the tracer experiment plots (b) for the four age classes.

The complexity values of the experiment plots are listed together with the results of the analysis of the fine roots and above ground biomass in Table 1.

**Table 1.** Main characteristics and vegetation parameters at the tracer experiment plots (n.a.= analysis was not carried out).

| Moraine age [years] | Normalized complexity | Complexity level | Vegetation Cover [%] | Slope [°] | Species Richness | RLD [km m$^{-3}$] | RD [kg m$^{-3}$] | SRL [m g$^{-3}$] | BM [kg m$^{-2}$] |
|---|---|---|---|---|---|---|---|---|---|
| 110 | 0.11 | low | 25 | 40 | 10 | 348.6 | 4.6 | 78.2 | n.a. |
| 110 | 0.32 | medium | 15 | 21 | 10 | 315.1 | 1.4 | 222.4 | n.a. |
| 110 | 0.34 | high | 55 | 25 | 13 | 150.6 | 0.8 | 129.8 | n.a. |
| 160 | 0.21 | low | 50 | 42 | 11 | 293.2 | 2.8 | 104.3 | 4.7 |
| 160 | 0.35 | medium | 20 | 35 | 11 | 201.4 | 1.3 | 172.3 | 5.7 |
| 160 | 0.54 | high | 75 | 23 | 13 | 165.9 | 3.0 | 58.1 | 3.9 |
| 4 900 | 0.58 | low | 90 | 23 | 21 | 939.8 | 10.2 | 93.0 | 5.5 |
| 4 900 | 0.60 | medium | 90 | 27 | 21 | 822.9 | 5.8 | 142.8 | 9.6 |
| 4 900 | 0.89 | high | 100 | 25 | 33 | 1235.16 | 7.8 | 161.4 | 8.7 |
| 13 500 | 0.56 | low | 85 | 27 | 25 | 654.2 | 9.5 | 68.3 | 5.0 |
| 13 500 | 0.76 | medium | 100 | 36 | 25 | 1197.2 | 10.4 | 121.3 | 3.6 |
| 13 500 | 0.86 | high | 80 | 37 | 27 | 608.7 | 6.0 | 111.8 | 4.6 |





## 3.2 Soil structure and texture

**Figure 4.** Evolution of bulk density (a), porosity (b), LOI (c) and soil texture (sand (d), silt (e), and clay (f) content) in the top 50 cm of the 4 moraines at the Griessfirn forefield in relation to vegetation complexity. Upper case letters above the boxplots indicate the results of the Tukey's HSD test among the age classes. Different letters denote significantly different mean values. Lower case letters denote the results among the vegetation complexity levels in each age class. n equals the number of observations in each individual box.

During the course of soil development over more than ten millennia the structural parameters and soil texture underwent significant changes (Figure 4). The displayed values show the distribution of the properties in the top 0-50 cm of the soil profile differentiated by vegetation complexity. For a depth differentiated presentation of the development of the soil properties see
5 Hartmann et al. (2020b).

With increasing moraine age the bulk density (Figure 4a) decreases distinctly. At the 110-year-old moraine, bulk density ranges mainly between 1.5 and 1.9 g cm$^{-3}$ and at the 160-year-old moraine fluctuates mainly around 1.5 g cm$^{-3}$. During the course





of soil development bulk density decreases to values lower than 1 g cm$^{-3}$ at the two older moraines, where the median values of the bulk density range between 0.5 and 1.0 g cm$^{-3}$.

Corresponding to the decrease in bulk density the porosity increases with increasing soil age (Figure 4b). The porosity at the youngest moraine fluctuates around 0.3 and at the 160-year-old moraine around 0.4. The median values at the two oldest

moraines are mainly in the range of 0.6 to 0.7, but also values lower 0.4 and higher than 0.8 were observed. At the 13 500-year-old moraine the medium complexity plot has porosity values in the same range as the other two plots, but the median value is distinctly lower (0.48).

The LOI (organic matter content) also increases with increasing soil age. No significant difference is observed between the two youngest moraines. The LOI at both moraines is mainly below 4 weight-% and the interquartile range (IQR) is small. The two

oldest moraines both show higher values and more variability. Median values at both moraines fluctuate around 10 weight-%, but also values up to 30 weight-% were occasionally observed at both moraines.

The soil texture also undergoes a significant change during the course of soil development. Whereas the youngest moraines are mainly composed of sand, silt is the dominant particle size at the oldest moraines. At the 110 and 160-year-old moraines, the sand fraction ranges mainly between 50 and 75 weight-% whereas at the two oldest moraine the median values of the sand

content are mainly below 25 weight-%, except for the plot with a low vegetation complexity at the 4 900-year-old moraine, where the median value of the observed sand fraction is still higher than 50 weight-%. However, the variability at the oldest moraines is very large. The silt content at the young moraines is mainly < 25 weight-% and variability is low. At the old moraines, the silt content is the highest with median values > 50 weight-%. No clear trend is visible for the development of the clay content. While the variability is comparatively high at the two old moraines and low a the two young moraines the clay

content lies generally between 12 and 25 weight-% without a strong tendency along the chronosequence. However, at the two old moraines occasionally higher clay amounts (compared to the young moraines) were observed.

We used the Tukey's HSD test to test whether the differences between the individual age classes are statistically significant. The test results of the Tukey's HSD test confirm the observation that with increasing moraine age the investigated soil properties change significantly. The observations at the two old moraines are always significantly different from the observations at

the two young moraines. The soil characteristics at the two young moraines, which are only 50 years apart in age, are not significantly different from each other, except for porosity. The two old moraines, which are more than 8 000 years apart in age, differ significantly only in their silt and clay content.

### 3.3 Vertical dye pattern analysis

The vertical flow pattern characteristics show no clear and consistent trend with respect to age, irrigation intensity or vegetation

complexity. This applies to both, the volume density profiles (Figure 5) and the surface area density profiles (Figure 6). To clarify if the stained area described by the volume density is made up of many small flow paths or few large ones the volume density has to be jointly interpreted with the surface area density.

**Figure 5.** Volume density profiles per age class, vegetation complexity and irrigation intensity. The volume density is the fraction of stained pixels, here color coded by flow path width (stained path width, SPW) and rock sizes. Infiltration depth was only determined when the dye pattern ended within the excavated profile (plot design after Hartmann et al. (2020a)).





**Figure 6.** Surface area density profiles per age class, vegetation complexity and irrigation intensity.

The patterns of the volume density and the surface area density profiles of the youngest moraine differ across the vegetation complexity levels. The results of the 110-year-old moraine can only be evaluated by taking into account various uncertainties (for more details see discussion, chapter 4.5). The low complexity plot shows a distinct reduction in tracer amount and stained path widths with depth. The infiltration depth is significantly smaller compared to the other complexity levels. However, the

5 results of the low complexity plot are based on only two observations, as the excavation had to be interrupted due to an unforeseen change in weather conditions. Despite protection, the rest of the plot collapsed due to a thunderstorm. At the medium complexity plot the volume density decreases with depth. The volume density in the top 25 cm is high and consists mainly of stained path widths >200 mm. The surface area density is also rather small and does not decrease much with depth. The





volume density and surface area density are highest at the lowest intensity. The high complexity plot has a high volume density with mostly stained path widths > 200 mm throughout the profile and a comparable low surface area density, which indicates a homogeneous tracer distribution. Whereas the plot with the high vegetation complexity shows an almost complete coloring of the soil, the dye profiles at the other two plots show a distinct reduction in tracer amount and stained path widths with depth.

The infiltration depth at the low complexity plot is smaller compared to the medium and high complexity plot, but a comparison is highly uncertain due to the reduced data base at the low complexity plot.

At the 160-year-old moraine no systematic trend with complexity level can be identified. All complexity levels have an almost complete staining of the topsoil in common, which mostly consists of stained path widths > 200 mm (Figure 5). At the low complexity plot infiltration depth, dye coverage, amount of flow path widths > 200 mm, and the surface area density decrease

step wise with increasing irrigation intensity. However, caution must be taken when evaluating the results of the low complexity plot due to a very large boulder that was located in the 40 mm h$^{-1}$ intensity subplot, which did not leave much soil to be analyzed (see Figure 7). At the medium complexity plot the dye coverage is very small. The profiles of volume density and surface area density show a strong reduction beyond 20 cm soil depth. The high complexity level has the biggest fraction of stained soil compared to the other levels. Broad flow paths (width > 200 mm) have the largest share of the dyed area. No

particular influence of irrigation intensity can be identified.

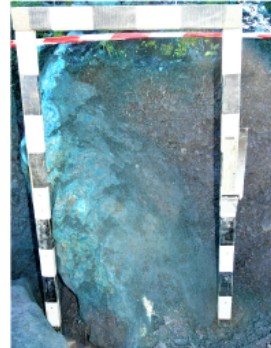
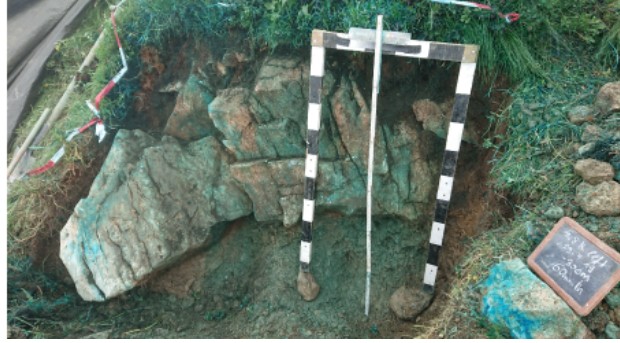

**Figure 7.** Left: Boulder at the low complexity plot at the 160-year-old moraine; Right: Boulder at the medium complexity plot at the 4 900-year-old moraine. Length of one black or white scale segment equals 10 cm.

At the 4 900-year-old moraine, the medium complexity plot can only be included in the comparison of the dye pattern profile by taking into account that a large boulder was located under the entire plot, which leads to a strong reduction in soil space (see Figure 7). At the low complexity plot the blue dye infiltrated deep into the soil, past the end of the excavated profile. The

blue pattern inversely mirrors the stone distribution for all three intensities, which indicates that stones in particular led to the interruption of the blue stained paths. The proportion of wide flow paths is gradually reduced while the dye coverage changes only slightly. The surface area density, on the other hand, is lower than at the other intensities (Figure 6), which suggests a larger area of stained soil for the low irrigation intensity. The high complexity plot shows no differences between the intensities,



but compared to the low complexity plot the volume density is lower.

The oldest moraine was the only one with no problems or uncertainties in the evaluation. The infiltration depth was greater than 1.0 m at all plots. The dye coverage is roughly the same at all plots, but decreases were more pronounced with depth at the medium complexity plot. At the low complexity plot the dye coverage, wide flow paths, but also the surface area density

5    increases continuously with increasing irrigation intensity.

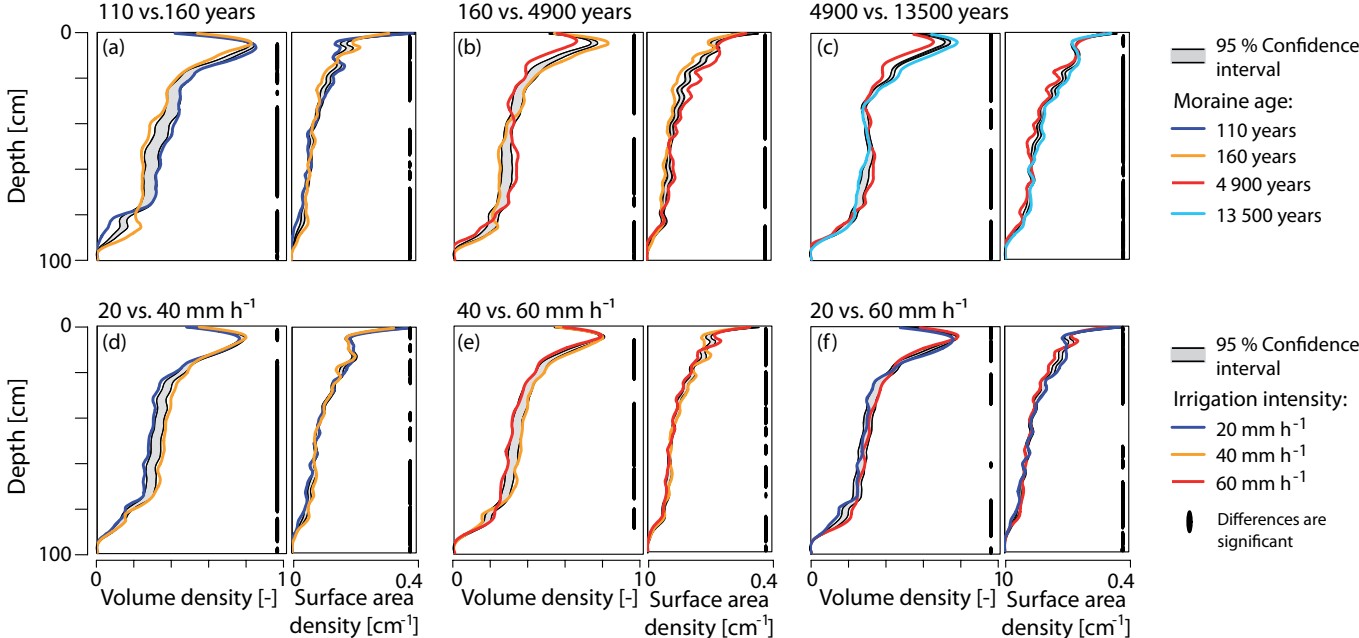

**Figure 8.** Differences of volume density and surface area density profiles with respect to (a-c) moraine age and (d-f) irrigation intensity. If the profile lines sit outside the grey-shaded confidence interval, the two profiles are considered to be significantly different. The parts of the depth profiles where this is the case are indicated by black vertical bars on the right of each plot. A BLR-test for differences in volume density profiles and surface area density profiles among the three irrigation intensities per age class is displayed in Figure A1.





**Figure 9.** BLR-test for differences in volume density profiles and surface area density profiles among the three complexity levels per age class. If the profile lines sit outside the grey-shaded confidence interval, the two profiles are considered to be significantly different. The parts of the depth profiles where this is the case are indicated by black vertical bars on the right of each plot.

To quantitatively asses the impact of age, irrigation intensity, and vegetation complexity on the dye pattern a statistical approach in form of a bootstrapped LOESS regression (BLR) was used. The approach is suitable for a comparison of two data sets with profile observations (Keith et al., 2016). The results of the BLR approach for a pair wise comparison of the averaged volume density and surface area density profiles are shown in Figure 8 and Figure 9. Due to the strong interference by large boulders (Figure 7), the low complexity subplot irrigated with 40 mm h$^{-1}$ at the 160-year-old moraine and the complete medium complexity plot at the 4 900-year-old moraine were excluded from this analysis. Next to the 95 % confidence interval of the 1000 LOESS regressions (bootstrap resampled out of the combination of both compared data sets) the LOESS regression





of both original data sets are shown. The differences between the two profiles are significant, if the LOESS regression curves sit outside the confidence interval. It would actually be sufficient to plot only one LOESS regression of the original data sets (Keith et al., 2016), but we plotted both. Based on the moraine age as the test variable we compared the sets of all volume density and surface area profiles per moraine age along the chronosequence. We find statistically significant differences in the

volume density and surface area density profiles among the adjacent age classes (Figure 8 a-c). Between the moraine age of 110 and 160 years the volume density profiles differ significantly below a depth of 10 cm, whereas for the surface area density profiles significant differences occur in the top 50 cm. The profiles of the 160 and 4 900-year-old moraine differ significantly except for a small area between 40 and 50 cm. For the volume density profiles comparison of the 4 900 and the 13 500-year-old moraine the difference between the profiles is significant except for a soil depth of 30 to 60 cm. While for the surface area

density profiles, the differences are significant in the top 60 cm.

For the comparison with regard to the irrigation intensity, all profiles of the four age classes were combined into three intensity classes. The pairwise comparisons of the volume density profiles of the three irrigation intensity data sets show no significant differences between the irrigation intensities of 40 and 60 mm h$^{-1}$ as well as between 20 and 60 mm h$^{-1}$ (Figure 8 e-f). Between 20 and 40 mm h$^{-1}$ (Figure 8 d) the profile lines are close to the boundaries of the confidence interval, but are located

outside of the interval at a depth of 20-80 cm. Significant differences only occur occasionally for the surface area density in the top 10-20 cm for all three comparisons. The profile comparison carried out individually for each age class, which does not average across the effects of the age differences, shows similar results (Figure A1). Significant differences only occur in the top 10-20 cm at the age of 110 years. At the 160-year-old moraine, differences are significant throughout the profile depth between the 20 and 40, as well as between 40 and 60 mm h$^{-1}$ and between the 20 and 40, as well as 20 and 60 mm h$^{-1}$ at the oldest

moraine.

The profile comparison with regard to the vegetation complexity was carried out individually for each age class (Figure 9), since the vegetation complexity gradients of the age classes are not comparable (Figure 3). The comparisons show that the volume density and surface area density profiles are significantly different across the complexity levels in each age class. At the young moraines, the profiles in the upper 10-20 cm lie within the confidence interval, which means that the differences are not

significant. In the deeper layers, the profiles are further apart and are no longer in the confidence interval, indicating significant differences. This is particularly evident at the 110-year-old moraine. At the 160-year-old moraine, significant differences occur in the volume density and surface area density profiles from a depth of 10 cm between the low and high complexity and the medium and high complexity. The comparison of the low and medium complexity shows almost no significant differences throughout the upper half of the soil profile. At the 4 900-year-old moraine, only the low and high complexity plots can be

compared with each other due to a large boulder at the medium complexity plot. The differences in the profiles are significant almost throughout the entire profile depth, except in a depth from 30-50 cm, where the profiles intersect.

At the oldest moraine, the profiles are comparatively closer to the confidence interval than in the majority of the other age classes, although here too the differences are classified as significant. When comparing the low and high complexity, the profiles of the volume density and the surface area density are within the confidence interval at a depth of 60-80 cm depth.

Comparing the low and high complexity, the volume density profiles are occasionally within the confidence interval at 0-20




and 60-80 cm. When comparing the low and medium complexity, far greater differences can be seen with only one identical section of the volume density profiles at 20 cm and a few more for the surface area density profiles.

## 3.4 Flow type classification

Using the information in the volume density profiles and the stained path widths to characterize flow types (Weiler, 2001) we found that over the millennia flow types transition from matrix flow to preferential flow (Figure 10 a).

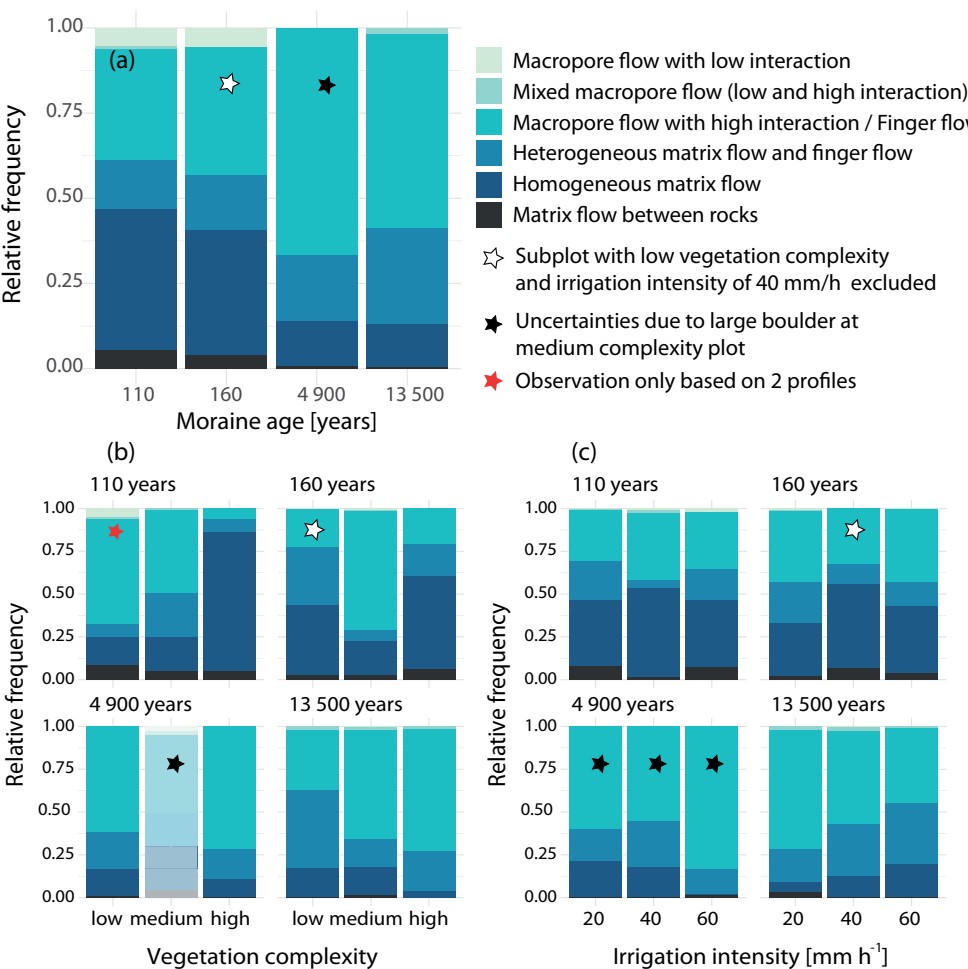

**Figure 10.** Relative frequency distribution of flow types (a) for the four moraine age classes, (b) differentiated by vegetation complexity for each moraine age, and (c) differentiated by irrigation intensity for each moraine age (c).

At the two youngest moraines homogeneous and heterogeneous matrix flow and finger flow are the predominant flow types. With increasing age of the moraine, the proportion of matrix flow decreases in favor of increased finger flow. At the two oldest moraine the dominant flow type class is 'Macropore flow with high interaction/ Finger flow'. The dominant flow type of this





joined flow type class (Hartmann et al., 2020a) is finger flow along small diameter flow paths, since no actual macropores were found during the field experiment.

The breakdown of the flow types for each age group depending on the vegetation complexity (Figure 10 b) shows that at the youngest moraine the homogeneous matrix flow was only dominant at the high complexity plot. With increasing complexity

level the proportion of matrix flow increases and the proportion of preferential flow paths in form of finger flow continuously decreases. At the 160-year-old moraine, there is no systematic influence of the vegetation complexity visible. It is striking that the plot with the medium vegetation complexity is strongly influenced by preferential flow paths in form of finger flow, whereas matrix flow plays a larger role at the low and high vegetation complexity. At the two old moraines, the importance of matrix flow decreases and the flow type class 'Macropore flow (high interaction)/finger flow' plays the dominant role. Again,

the dominant flow type of this joined flow type class is finger flow with small diameter flow paths, since also at these age classes no flow in macropores was observed during the field experiment. At the 4 900-year-old moraine, finger flow is almost equally dominant at the low and high vegetation complexity. At the medium vegetation complexity level, homogeneous and heterogeneous matrix flow play a more important role, but the results of this plot are highly uncertain due to the large boulder. At the 13 500-year-old moraine the proportion of preferential flow increases continuously with increasing complexity, whereas

the proportion of heterogeneous matrix flow decreases.

With regard to the irrigation intensity no consistent impact on the flow type distribution across the millennia can be identified (Figure 10 c). At the 110-year-old moraine the two dominant flow types show an almost equal distribution at all irrigation intensities. At the 160-year-old moraine the flow type distributions are also quite similar at all irrigation intensities. At an intensity of 20 mm h$^{-1}$ the the proportion of heterogeneous matrix flow is slightly greater and the proportion of homogeneous matrix

flow is correspondingly slightly lower compared to the other two irrigation intensities. At the 4 900 year old moraine finger flow is more dominant at the irrigation intensity of 60 mm h$^{-1}$. The proportion of homogeneous matrix flow decreases with increasing irrigation intensity. The oldest moraine is the only age class where the flow type frequency distribution shows a trend with irrigation intensity. Finger flow is mainly present at the low intensity. With increasing irrigation intensity, the frequency of finger flow gradually decreases and the importance of both homogeneous and heterogeneous matrix flow increases.

## 4    Discussion

### 4.1    Evolution of soil structure and texture

During the course of soil development of more then ten millennia we see significant changes in soil structural parameters and soil texture. The change in soil texture with a decrease in the sand fraction and an increase in the silt and clay fraction has also been observed by (Dümig et al. (2011), He and Tang (2008), Douglass and Bockheim (2006)). Our previous study on moraines

of the Stein glacier, with soils which developed on siliceous parent material, showed similar results with a less pronounced reduction in sand content in the upper layers and a less pronounced increase in silt and clay fraction (Hartmann et al., 2020a). This break-down in particle sizes without changing the particle mineralogy (Ellis, 1992) is caused by physical weathering, i.e. high fluctuations between day and night temperatures and freezing cycles (Birse, 1980). The higher sand content at the



siliceous parent material is common for its partly granitic origin (Musso et al., 2019). The calcareous soil containing schist at the Griessfirn forefield of this study is also prone to chemical weathering in form of carbonate dissolution, which results in a further enrichment of fine particles (Musso et al. (2019), Fleming et al. (1963)).

The trends with age in porosity, bulk density and LOI have also been found by Crocker and Major (1955), He and Tang (2008) or Vilmundardóttir et al. (2014) for a period of 100-200 years of soil development and in our previous study (Hartmann et al., 2020a) for a period of 10 000 years. A significant decrease in bulk density and an increase in porosity through the millennia can be attributed to the change in the grain size distribution as well as to the ongoing vegetation development and soil biological activities with an increase in root activities and organic matter accumulation. The exceptionally high porosity at the oldest moraines occurs primarily in the top soil and is caused by the high organic matter content (Nyberg (1995), Carey et al. (2007)).

The amount of organic matter content found in the top soil at the oldest moraine was much lower than at the oldest moraine (age of 10 000 yr) in our previous study in the siliceous geology (Hartmann et al., 2020a). The observed values there ranged from 6 to 87 weight-% with a median value at 38 weight-%, which is much higher than the median value of ∼21 weight-% found during this study at the calcareous forefield. At the oldest moraine of siliceous parent material, the organic layer ranged in thickness from a few centimeters to 30 cm. The pH-value of ∼4 found at this moraine is very low (Musso et al., 2019). Due to the different parent material the pH-values at both chronosequences are extremely different. At the calcareous forefield of the Griessfirn, pH-values range between 7 and 8 over the chronosequence and at the siliceous forefield of the Stein glacier the pH-value decreases with age (Musso et al., 2019). The extremely low pH-value at the siliceous forefield likely restricts biotic decomposition and favors the building of a thick organic top layer. The low pH-value results from acidic weathering of the parent material (Musso et al., 2019) which attracts plants, such as alpine rose (*Rhododendron ferrugineum*) or blueberry (*Vaccinium myrtillus, Vaccinum uliginosum*) (Maier et al., 2020) that release root exudates which decrease the pH even further (Ellenberg, 2010). At the oldest moraines in the calcareous forefield, soil organism such as earthworms and snails were frequently present but this was not the case at the siliceous forefield, which is a result of the acidic milieu of the organic top layer at the siliceous forefield (Schmera and Baur (2014), Rożen et al. (2013)). A detailed comparison of the soil texture and soil structure data at both forefields can be found in Hartmann et al. (2020b).

## 4.2 Evolution of flow paths

The BLR-analysis showed that the difference in dye patterns as a result of the subsurface flow paths is statistically significant from one age to the next, which indicates an impact of age on the Brilliant Blue dye staining pattern. The adapted (Hartmann et al., 2020a) flow type classification by Weiler (2001) was used to derive prevailing flow types based on the staining patterns. We found that the two youngest moraines have nearly similar flow type distributions. The same is true for the two oldest moraines (Figure 10 a). Whereas homogeneous matrix flow plays a strong role at the young moraines, its influence declines with age with an increase in heterogeneous matrix flow and finger flow.

At the two youngest moraines, distinctly more surface runoff was observed during the irrigation experiments than at the older moraines. Unfortunately, the amount of surface runoff could not be quantified. Individual attempts to measure surface runoff at the 4 900-year-old moraine led to estimates of a maximum surface runoff of 100-200 ml per subplot (total irrigation amount





20 L). From purely visual observation during irrigation it seemed like the amount of surface runoff increased with irrigation intensity at the younger moraines, which might be due to the occurrence of structural sealing and is discussed in the following section. However, when not affected by structural sealing a homogeneous and deep infiltration was observed (Figure 5). The pore space of coarse-textured, unsorted, sandy soil is mainly made out of large pores, which provides only a low water retention

capacity and leads mostly to a fast downward transport of water (Hartmann et al., 2020b). The root system has a low density (Table 1, Greinwald et al. (2021a)) at the early stages of vegetation succession and does not impact the water transport. The low vegetation cover also does not inhibit infiltration and the water can infiltrate deep into the soil. Only larger stones and occasional clay lenses (the size of a few centimeters) or material heterogeneities influence the water transport and create heterogeneous matrix flow.

At the two oldest moraines, the vegetation cover was dense and probably has a high interception storage capacity (albeit reduced by vegetation trimming). The fine textured soil with a higher porosity (and higher proportion of fine pores) and lower bulk density has a higher water retention capacity. The soil is heterogeneous with a higher organic matter content in the upper layer and depth-gradients in porosity (decreasing) and bulk density (increasing) (Hartmann et al., 2020b). The root system is dense with most of the root mass (> 90 %) located in the uppermost 30 cm (Greinwald et al., 2021a). At both moraines

deep infiltration, but almost no surface runoff was observed. The image analysis has shown that finger flow dominates, with fingers already induced at the soil surface or within the upper 20 cm. Thus, water infiltrated heterogeneously and/or water transport was affected by the properties of the upper soil layer. Hydrophobic properties of the soil are often considered to cause finger-like flow paths (Wallach and Jortzick (2008), Dekker and Ritsema (2000), Blume et al. (2008) Hardie et al. (2011)). The humid and cool climate of former glacial areas leads to a slow decomposition of biomass and thus to an accumulation

of hydrophobic compounds (Doerr et al., 2000), which are released during the decay of litter (Reeder and Jurgensen, 1979) or by root activity (Doerr et al., 1998). These hydrophobic compounds coat soil particles or are deposited in the pore space. But also observed material heterogeneities such as gravel and sand patches likely facilitated the formation of finger flow paths. The high frequency of finger-like flow paths was also found during our previous study at the 160 and 3 000-year-old moraines at the siliceous forefield, both of which had a dense vegetation coverage. Thus, we again assume that hydrophobicity of the

organic top layer has a big impact on infiltration and the initiation of unstable flow. In general, the flow type distribution at the first three age classes are quite similar at both forefields and thus both geologies. At the siliceous forefield the oldest moraine is dominated by macropore flow, whereas at the calcareous forefield the, by Hartmann et al. (2020a) newly introduced, joined class of finger flow and macropore flow (high interaction) is prevailing. Based on field observation it has to be stated that in this case finger flow is the prevailing flow type, since no macropores were observed.

At the oldest moraine at the siliceous forefield, the thick organic top layer held back most of the water and only a little percolated deeper into the soil via macropores. The organic layer at the calcareous forefield was significantly less thick and water was able to percolate deep (> 1 m) into the soil via finger flow.





### 4.3 Impact of vegetation complexity

**Soil structure and texture**

When considering the relation of vegetation complexity to the soil parameters, the scale differences in the vegetation complexity estimation between the age classes must be taken into account. The complexity levels of the soil sampling plots at the 160,

4 900, and 13 500-year-old moraines were determined on a large scale (> 20 m$^2$), whereas at the 110-year-old moraine the complexity levels were determined on a smaller scale (1.5 m$^2$). At the 110-year-old moraine, a trend in soil parameters can be seen with increasing vegetation complexity. The median values of LOI increase continuously. However, the differences are only very small and also not statistically significant. At the same time, the porosity increases and the bulk density decreases with increasing complexity. The statistical significant differences in bulk density, porosity, and clay content between the low

and medium complexity plot correspond to the quantitative differences in vegetation complexity (Figure 3a), which are high between the low and medium/high and smaller between the medium and high complexity levels.

At the other age classes, no statistically significant differences in the soil parameters of the different complexity levels could be determined. The 160-year-old moraine shows no clear differences in soil properties between the complexity levels. Despite the fact that a clear difference in complexity value between the low complexity and the other two levels exists (Figure 3a). At the

4 900-year-old moraine, the median values of porosity, LOI, silt content, and sand content differ clearly (although not statistically significant) between the low complexity and the two other levels, which corresponds with the differences in complexity values of the three levels. At the 13 500-year-old moraine, no systematic differences in median values can be seen, which is in accordance with the very small differences between the three complexity levels (Figure 3a).

While the number of two sampling locations per plot is justifiable for the small plots at the 110-year-old moraine, it must be

viewed critically for the larger sized plots at the older moraines. In the case of the 160-year-old moraine, it can be assumed that due to the highly heterogeneous and patchy vegetation distribution, the small-scale vegetation complexities within the plot might also be very heterogeneous and partly distinctly different from the large-scale complexity. Therefore, only two sampling locations are not representative for the plot size and the connection between the observed soil properties and the large-scale complexity level is highly uncertain. At the two older moraines, however, the vegetation cover is dense and the distribution is

more homogeneous, thus small-scale differences in vegetation complexity are likely not very pronounced. Which could possibly be the reason that at the older moraines the relations of the differences in the vegetation complexities are also quite similar to the differences in the soil properties.

**Subsurface flow paths**

The dye profile comparison with regard to the vegetation complexity using the BLR-approach showed that at each age class the volume density and surface area density profiles of the complexity levels are significantly different (Figure 9). At the 110-year-old moraine, the smaller differences between the profiles at the low and medium complexity and larger differences between the medium and high complexity, does not reflect the differences between the individual complexity values of the three complexity





levels (see Figure 3b). At the 160-year-old-moraine, the qualitative differences in the profiles of the complexity levels are also not equivalent to the quantitative differences in complexity values (Figure 3b). Whereas the complexity values at the three levels differ almost equally, there are strong differences in the profiles between the medium and high as well as low and high complexity plots and smaller differences between the low and medium complexity plots.

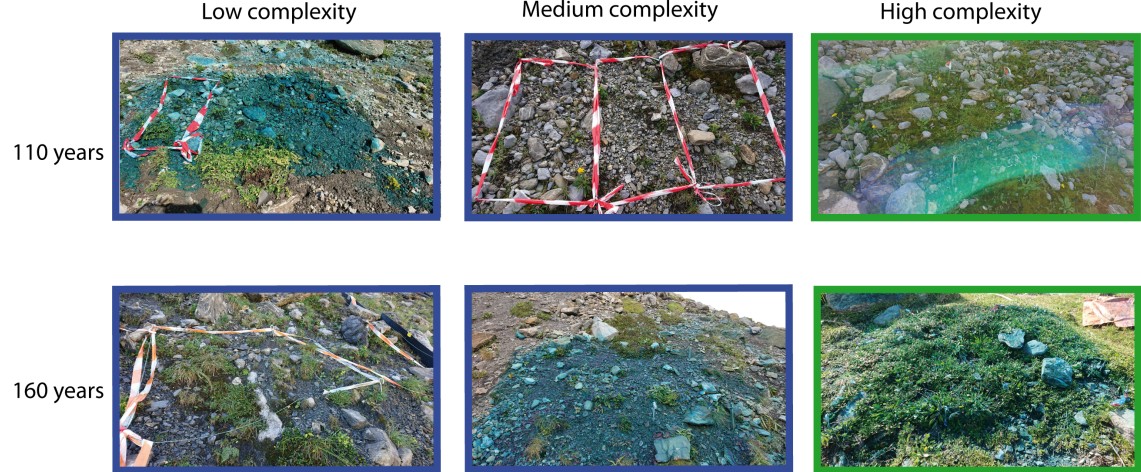

**Figure 11.** Surface of the three plots at the 110 and 160-year old moraine. Frame color shows plots with a similar degree of vegetation coverage.Photograph of the high complexity plot at 110-year old moraine is disrupted by lens flares due to backlighting.

At the 110-year-old moraine, flow types change along with vegetation complexity (Figure 10 b). With increasing complexity, the proportion of preferential flow paths continuously decreases. This is not the case at the 160-year-old moraine. The proportion of preferential flow paths is high at the medium complexity plot and low at the low and high complexity plot. At both age classes, the high complexity plots show a deep staining of the soil profile along with a generally high dye coverage (Figure 5) and are dominated mainly by a homogeneous matrix flow (Figure 10 b). The deep infiltration and homogeneous water trans-
port are probably less linked to vegetation complexity and more to the vegetation coverage. Both high complexity plots show a high degree of vegetation coverage (> 50 %, Table 1), which is evenly distributed over the entire plot area (Figure 11). In comparison, the low and medium complexity plots at both moraines (Figure 11) have a similar surface coverage with only a few single vegetation patches between gravel and small stones. Only the vegetation coverage at the low complexity plot at the 160-year-old moraine is also around 50 %, but the majority of the vegetation is here concentrated on the left subplot, which
was irrigated with 20 mm h$^{-1}$. This subplot shows mainly the same flow patterns as the two high complexity plots. Only at this subplot a deep infiltration and higher proportion of wider flow paths was observed (Figure 5). Thus, the large proportion of heterogeneous matrix flow at the low complexity plot (Figure 10 b) results mainly from the subplot irrigated with 20 mm h$^{-1}$. In comparison, the subplot irrigated with 60 mm h$^{-1}$ shows a distinctly lower infiltration depth and a smaller amount of wider flow paths. The subplot with 40 mm h$^{-1}$ irrigation could not be compared due to the disturbance by the large boulder.
On the barely covered plots at both young moraines, the water infiltrated homogeneously in the top 10 cm, but there was hardly





any staining in the soil below this depth (Figure 5). Together with the observed surface runoff, which started a few minutes after the start of the irrigation, it can be assumed that after initial infiltration, the infiltration capacity of the soil surface decreased rapidly and thus overland flow was induced. This could have been caused by ensuing structural sealing of the soil surface due to irrigation with high intensities. Loamy to sandy soils with low organic matter content are prone to building structural seals

resulting in a reduction in near-surface porosity and unsaturated hydraulic conductivity (Armenise et al., 2018). The sealing of the soil is caused by the disruption of the soil surface structure and wash-in of released fine particles that leads to clogging of near-surface pores (Assouline, 2004). We assume that this is the reason for the observed rather small infiltration depths at these coarse textured soils with high hydraulic conductivities near saturation and a small water holding capacity (Hartmann et al., 2020b). Structural sealing was not observed in our previous study, but there only the youngest moraine had a sparse vegetation

coverage and the silt and clay content was lower.

At the 4 900-year-old moraine, the complexity values between the low and medium complexity levels differ only very slightly, but the two are distinctly different from the high complexity level (Figure 3b). Since the results of the irrigation experiments at the medium complexity plot are very uncertain due to a large boulder, an evaluation can only be made for the low and high complexity levels. The frequency of preferential flow is higher at the high complexity plot. However, the difference in prefer-

ential flow frequency between the two complexities is not as pronounced as the difference in their complexity values. At the 13500-year-old moraine, the proportion of preferential flow paths also increases with increasing complexity. The differences in flow types qualitatively correspond to the quantitative differences in the complexity values (Figure 3). It can therefore be assumed that the vegetation complexity, which combines above-ground and underground vegetation characteristics (together with further functional vegetation aspects), can be related to the formation of preferential flow paths.

## 4.4   Impact of irrigation intensity

In previous studies at the catchment scale, an increase in preferential flow paths with increasing rainfall intensity has been observed by evaluating the sequence of water content responses in different depths (Lin and Zhou (2008); Wiekenkamp et al. (2016), Demand et al. (2019)). The observations from our dye tracer experiments at the plot scale, however, did not show this

tendency. Also the BLR-test between the volume density profiles of the three irrigation intensities averaged over the four age classes (Figure 8 e-f) and per age class (Figure A1) showed mostly no significant differences in the volume density profiles. An exception is the oldest moraine, here the BLR-test showed significant differences in the lower part of the volume density profiles when comparing the lowest intensity of 20 mm h$^{-1}$ with the two higher intensities of 40 and 60 mm h$^{-1}$ (Figure A1). At this age class, the flow path classification showed a systematic shift from mainly preferential flow paths in form of finger

flow to a rather heterogeneous to homogeneous matrix flow with increasing irrigation intensity. Only at the 4 900-year-old moraine a slight tendency to a higher frequency of preferential flow paths with increasing intensity was observed.

Contradictory findings in the occurring of preferential flow depending on specific factors are quite common. Cichota et al. (2016), for example, observed an increase in preferential flow paths with increasing irrigation intensity under soil moisture conditions near field capacity (comparing rainfall events of 5 mm h$^{-1}$ and 20 mm h$^{-1}$). Wu et al. (2015), however, found,





based on Brilliant Blue experiments, a decrease in preferential flow paths at higher intensities (comparing rainfall events of 50 mm h$^{-1}$, 100 mm h$^{-1}$, and 150 mm h$^{-1}$) when the initial soil water content was high. As we only irrigated each plot once, our results cannot be evaluated in relation to different antecedent soil moisture conditions. However, soils were generally quite wet during the entire field campaign as the total rainfall amount in this region is quite high and hence the soils never

really dry out. The high number of factors influencing preferential flow paths (e.g. initial soil water content, soil texture, soil hydrophobicity, etc) and different experimental designs complicate the comparison of different studies.

### 4.5    Uncertainties

Apart from the reduced data base at the low complexity plot at the 110-year-old moraine and the uncertainties due to large boulders at the 160 and 4 900-year-old moraines, some further uncertainties need to be discussed. In general, the dark gray soil

color at the 110 and 160-year-old moraine made the color detection of the tracer difficult. In addition, at the high complexity-plot of the 110-year-old moraine, a set up of suitable lighting conditions was difficult due to stormy weather conditions. As a result, the lighting of the photographs was very unfavorable for the image analysis. Even during the profile excavation in the field it was not possible to determine with great certainty whether the dark colored, wet soil was stained or not. Thus, blue stains on larger stones along the profile depth were considered as an indicator for the validity of the observed dye tracer pattern,

which shows an almost complete coloring of the soil (Figure 5).

We further assume that the irrigation with the hand-operated sprayer, which had to be guided close to the soil surface due to strong winds, in part led to a high force of application and promoted structural sealing at the 110 and 160-year-old moraines. At both moraines, deep infiltration was often found at the outer boundaries of the bare plots and at the transition boundaries between the individually irrigated subplots. We assume that this is caused by lateral runoff onto the soil surface that was

protected by the tarpaulin and not affected by the irrigation. This observation suggests that a more homogeneous and deep transport of the water can take place in this quite homogeneous and unsorted material (Hartmann et al., 2020b), if the surface is not influenced by particle displacement. Thus, it is assumed that the proportion of preferential flow paths at the young moraines is generally overestimated and homogeneous to heterogeneous matrix flow are the dominant flow types under natural rainfall conditions. The use of the tarpaulin to cover the non irrigated adjacent subplots and the irrigation method with the

hand-operated sprayer are therefore limitations of the experimental setup.

### 5    Conclusions

Based on Brilliant Blue dye experiments on a glacial chronosequence developed on calcareous parent material, we found that subsurface flow paths change with age; from homogeneous gravity driven matrix flow in sandy, coarse-grained, loose soils with low root density at the young moraines to a heterogeneous matrix and finger flow in silty, layered soils with dense vegetation

cover and high root density at the old moraines. This shows that the influence of preferential flow paths increases with soil age. Preferential flow paths were mainly controlled by soil surface characteristics such as vegetation patches, micro-topography and hydrophobicity and in case of the oldest moraine also by vegetation complexity. Pronounced changes in soil texture and





an ongoing accumulation of organic matter due to vegetation succession, which also affects bulk density and porosity, was observed. Regarding soil texture the biggest changes occurred in the sand and silt fraction.When infiltration was not impaired by structural sealing, water percolated deep into the soil (> 1 m) at all four age classes.

Independent of age, only at the oldest moraine the variation in irrigation intensity between 20 and 60 mm h$^{-1}$ showed an

influence towards less preferential flow paths with increasing intensity. However, the selected intensities are rather high and it remains unknown whether there is an impact at intensities below 20 mm h$^{-1}$.

We saw a direct relationship between vegetation complexity and subsurface flow paths at the old moraines and a relationship of vegetation complexity and soil properties at the 110, 4 900, and 13 500-year-old moraines. At the oldest moraines a higher vegetation complexity was related to more preferential flow in form of finger flow. Possibly due to limited data points and related

uncertainties we were not able to generalize this across all sites. However, we still suggest that a more in-depth consideration of vegetation characteristics beyond coverage and land use types will provide useful insights for hydrological process research. Our findings deliver important insights on how landscape co-evolution affects hydrological processes in transient alpine landscapes, where glacial retreat is accelerating and thus more and more hillslopes are freed of ice and weathering, erosion and plant succession are initiated. The here provided data and observations can also help to improve the handling of hydrologic

processes and their role within the feedback cycle of the hydro-pedo-geomorphological system when it comes to soil and landscape evolution modeling.

*Code and data availability.* The soil structure and texture data is described in detail in Hartmann et al. (2020b). It is available at the online repository of the German Research Centre for Geosciences (GFZ, Hartmann et al. (2020c)) and can be accessed via the DOI: https://doi.org/10.5880/GFZ.4.4.2020.004

The brilliant blue images can be obtained from the authors upon request.





## Appendix A

**Figure A1.** BLR-test for differences in volume density profiles and surface area density profiles among the three irrigation intensities per age class. If the profile lines sit outside the grey-shaded confidence interval, the two profiles are considered to be significantly different. The parts of the depth profiles where this is the case are indicated by black vertical bars on the right of each plot.

*Author contributions.* AH conducted the tracer experiments, soil sampling, and the laboratory analysis. KG conducted the estimation of the vegetation complexities. AH prepared the images and performed the analysis. MW and TB were involved in planning the fieldwork. AH prepared the manuscript with contributions from all co-authors.



*Competing interests.* The authors have the following competing interests: Markus Weiler is editor of HESS. Theresa Blume is chief-executive editor and editor of HESS.

*Acknowledgements.* This work is funded by the German Research Foundation (DFG) and the Swiss National Science Foundation (SNF) within the DFG-SNF-project Hillscape (Hillslope Chronosequence and Process Evolution). We thank Nina Zahn, Wibke Richter, Louisa Kanis, Peter Grosse and Carlo Seehaus for their persevering assistance in the field. We also thank the Canton Uri, the community Unterschächen and Korporation Uri for permission to conduct the experiments. Many thanks to Christine, Franz and Matthias Stadler at Chammlialp for
5    their support and kind hospitality.



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
