# Peer review of "The impact of soil development, rainfall intensity and vegetation complexity on subsurface flow paths along a glacial chronosequence of 10 millennia"

_Hydrology and Earth System Sciences, 2021_

## Author Comment (AC1)

**Response to Reviewer comments**

**Response to Reviewer 1**

**General Comments**

Having conducted research on subsurface flow, in particular preferential flow, for many years, I was very interested to review this paper. I always like to see such investigations that include detailed and well-planned field work. The senior author is commended for this detailed staining field work, but the linkages with other aspects of the study are weak and almost ad-hoc. Also, the planning of this study could have been improved (see comments that follow).

In reading through this paper, I found many similarities with the other two papers recently published by these authors (i.e., Hartmann et al., 2020a,b), one in this same journal and another in Earth System Sci. Data. We really need to be conscious about not republishing similar material; I am sure that was not the intent here, but this paper does come across as having at least some evidence of this issue. I fully realize that new infiltration studies were conducted as part of the paper I reviewed, but the authors did not do a good job in showing how these new findings helped address the key stated issue of the feedback cycle of the hydro-pedo-geomorphological system associated with these glacial chronosequences. As such, the current manuscript is overly wordy and repetitive, and poorly links the dye experiment with the vegetation complexes and soil properties, partly because of the mismatch of scales. Thus, I do not see this paper as acceptable in HESS and recommend that the authors carve out the new piece on the dye staining experiment and submit this elsewhere as a short note.

**Response to General Comments**

The authors would like to thank the reviewer for spending her/his time on this review.

We regret that the reviewer was left with the impression that the study presented here as a follow-on to our previous study on different bedrock material does not provide sufficient novelty. Nevertheless, we are convinced that there is value in our study and will reply to both the general assessment as well as the specific comments below.

First of all, the similarity of the experiments presented here with our study previously published in HESS (Hartmann et al., 2020a) was fully intentional. To our knowledge Hartmann et al., 2020a was the first systematic study of the co-evolution of flow paths and soil properties along a chronosequence of hillslopes in a glacial forefield developed on siliceous parent material. The replication of such a study at a site with differing parent material is of great importance for the verification and generalization of the gained knowledge. In our opinion, this applies above all to the complex interplay within the hydro-pedo-geomorphological system, where the parent material has a significant influence. It was therefore very important for us to make it clear within the manuscript that this is a follow on study on calcareous parent material, which therefore necessarily has corresponding similarities to the study on siliceous parent material. For this reason, we again chose HESS for this submission, in order to facilitate the readers' access to both parts of the study. It should also be noted that two slightly different angles were used in the two studies. In addition to the development of the preferential flow paths with moraines age (investigated in both studies), the development of flow paths depending on irrigation amount was the second focus in the first study, while the current study looked at the dependence on the irrigation intensity.

In the specific comments, the reviewer adds the justified comment that the results of both field campaigns could have been processed in a joint paper. However, such a single publication would have meant to cut down on the details of the results significantly and focusing only on those which are directly

comparable. But given that these types of systematic studies are exceedingly rare we felt that the community was better served by providing the full picture of results found at each of the study sites which necessarily meant to split it into two separate submissions to avoid an excessively long publication.

Furthermore, within this study we also set the focus on the influence of vegetation complexity. This was only considered tangentially in the previous study. The reviewer rightly argues that the different scales of the vegetation complexity for soil sampling and flow path determination appear poorly designed and make a concrete evaluation with regard to this aspect difficult. Our study is an interdisciplinary study in which, due to the complexity of the hydro-pedo-geomorphological system, we not only rely on our own expertise, but also work together with other disciplines such as geobotany, geography and geomorphology. While the advantage of this collaboration is the interdisciplinary exchange of knowledge, there is also the fact that tradeoffs have to be made when designing experiments, partly due to competing interests.
Even if we would have used other aspects from hydrologist point of view for the experimental design and the plot selection, it was of interest to us to use the new data set to investigate whether a common geobotanical proxy for plot selection (combining surface and subsurface vegetation characteristics such as coverage, root type, etc.) has an influence on our observations of soil properties and flow paths. Even though this specific analysis did not produce any outstanding findings, the fact that we did not find significant effects of vegetation is also an important result and we see the discussion of this issue as fruitful, especially in the context of the need for interdisciplinary studies of the hydrological system.
In addition, the reviewer correctly notes that the soil data have already been published in Hartmann et al., 2020b. However, this publication is a pure data publication that only supplies the data set to the community and does not contain any analysis and evaluation/interpretations of the results. The first time that this was done was within this manuscript. Furthermore, the data publication shows a depth-differentiated representation of the data set at each age class, while here we represent a depth-averaged breakdown with regard to the vegetation complexity.

We furthermore take the reviewers' comments on the wordiness very seriously and will definitely address this issue in a revised manuscript.

Specific Comments
The Abstract of this paper needs to focus on the key findings. Currently it focuses most on methodology.

We agree with this statement and will improve the abstract accordingly.

Section 1 – Introduction
Pg. 2, L. 12-13: The statement about preferential flow occurring more often at high rainfall intensities really depends on the environment you are working in and the type of preferential flow pathways that exist. In drylands where Hortonian overland flow dominates, few opportunities for subsurface preferential flow may exist, unless large cracks appear on the land surface that intercept surface runoff and route this into the subsurface.

We see the reviewers point and will clarify the sentence in that we are actually talking only about humid areas. Our and the referenced studies were restricted to humid regions.

Pg. 2, L. 19-28: This entire paragraph could simply be restated that most past chronosequence studies where soil development has been investigated are 2-dimensional – i.e., they only examine pedons, not 3-D flow paths, which is basically what was done in this study.

We would prefer to provide the more detailed information on this different aspects covered by previous studies. We are furthermore slightly reluctant to claim that we studied 3-D flow paths as due to the way we excavated the soil profiles the focus is more on a 2-D evaluation, at best quasi-3-D.

Pg. 2-3 last paragraph on pg. 2: You need a transition to this paragraph. Importantly, the reference to your previous and very similar paper in HESS (Hartmann et al., 2020a) summarizes that "observed flow types changed from a rather homogeneous rapid matrix flow in coarse material at the youngest moraine to a mainly finger flow at the medium-age moraines". Given the small plots and the methods for establishing 'horizontal connectivity' among excavated slices of the profile (referenced in a difficult to access thesis), I am not convinced of the generalization you make – particularly the assumption that mainly finger flow occurs in 'medium-age' moraines. Given the heterogeneity that occurs in moraine material, there certainly could have been some preferential flow paths that were missed (at larger scales). Even in compact glacial tills, preferential flow can occur in glaciotectonic fractures and desiccation fractures. It seems to me that while your inferences related to vertical preferential flow are quite good, those related to what you call horizontal flow are rather suspect. I fully realize that such 'horizontal' pathways are difficult to quantify, but here you stress these differences. I really think you could have combined these two papers as they are closely related; this would have reduced the repetitive material in the two papers and made for a much more comprehensive story.

We would like to clarify that we focused on vertical flow paths in both studies. We never made any references to horizontal pathways and it is also not mentioned in the paragraph addressed by the comment. In addition, we are not sure we understand the reviewer's point here as we did find preferential flow at the medium age moraines. Does the comment refer to the fact that we did not identify fractures or macropores but finger flow as the dominant feature? Our experimental design was aimed at covering the expected spatial heterogeneity by repeating the experiments in three different locations on each moraine. As it is nevertheless possible that fractures were present elsewhere on the moraines we refrain from stating that fractures are non-existent. However, extensive excavations and core profiles at additional locations on the same moraines as part of other experiments of this interdisciplinary effort did not reveal anything pointing at the importance of fractures. Desiccation fractures are unlikely due to the high soil moisture content throughout the study.

Pg. 3, L. 5-18: Here you spend an entire paragraph trying to justify why this paper is different from Hartmann et al. (2020a). In the previous paper, you also discuss vegetation effects, and I would pose the question: how is it possible not to consider vegetation (particularly below ground biomass) in discussions related to subsurface flow pathways? And the issue of irrigation intensity is not generalizable, as noted before.

That is correct. We made some effort to clarify to the reader that this is a follow up study which is in some aspects similar to the first study, but also has its unique features. In the previous study we also mentioned vegetation effects, but covered the vegetation complexity only tangentially. As already mentioned in the response to the general comment, it was of interest to us to investigate the impact of a vegetation proxy that combines both surface and subsurface vegetation characteristics in more detail,

since subsurface vegetation characteristics are often neglected when investigating subsurface flow paths in relation to vegetation coverage (Bachmaier et al., 2009, Stumpp and Maloszewski, 2010, Kan et al., 2020).

Even though the impact of irrigation intensity might be not generalizable, it surely is of interest for a specific landscape, environment, etc.

Pg. 3, L. 19-23: This material belongs in the methods section.

We believe that it is appropriate to briefly mention the applied methods in the introduction. But we agree to shorten this methodological description to a single sentence: "We used a statistical method specifically designed for the comparison of soil profiles (Keith et al., 2016) to assess the statistical significance of the differences across the observed dye profiles."

Pg. 3, L. 24-28: Not a well-articulated objective statement if that was what this was intended to be.

This paragraph was not intended to be the objective statement but was meant to provide background on the interdisciplinary study that the dye tracer experiments were part of. To clarify we will rephrase and move this section further up in the introduction.

Section 2.2: This entire section is very poorly connected with the stated objective related to subsurface flow paths. It is not clear why the 'structural vegetation complexity measure' was adopted, nor how this is linked to subsurface flow paths. The second paragraph begins with mentioning soil sampling, but then reverts to vegetation surveys; this is very disorganized and does not connect with subsurface flow. Many issues that seem not relevant to the discussion of subsurface flow are reported here, leaving me to wonder if this latter study (the one I reviewed) was an afterthought.

We agree that this section might have been too heavily focused on explaining the details of the vegetation complexity measure which might be distracting from the main goals of the study. We will completely revise this section to make it more easily accessible and to improve the logical reasoning behind the use of this measure as well as the logical flow of the text describing the sampling and surveying.

However, we do assume that a strong link between vegetation, soil properties, and subsurface flow paths exists as it was also previously shown that root activities and vegetation composition have an impact on weathering and soil water transport (Ivory et al., 2014; Amundson et al., 2007). The measure of vegetation complexity combines both, surface characteristics (vegetation coverage) and functional aspects of the vegetation in one measure. As it includes habitat composition and underground plant traits we hypothesized that it would be helpful in the framework of our study.

Section 2.3: It appears this dye study was conducted a year after the study reported in Hartmann et al. (2020a). However, the same plot design was used (subplots were 0.5 m x 1 m) and this raises the concerns I mentioned previously about difficulties in assessing horizontal preferential flow paths (especially across larger scales). These scale issues also affect other flow pathways, and this can be related to the variable infiltration rates used; the pathways that emerge may thus not be representative of larger scale behavior. Finally, no shortcomings of the Brilliant Blue dye methodology related to soil flow pathways was mentioned – this has been reported in numerous studies. At least this needs to be mentioned.

We have again to make clear that we only used this method to analyze vertical preferential flow paths. We never stated that this method would provide us insights on horizontal preferential flow paths, nor did we draw any conclusion on horizontal preferential flow paths. Our field observations and further measurements (including vertical hydraulic conductivity, retention curves) indicate that in the fast draining soils the vertical water transport was the dominant process.
Regarding the draw backs of the well-established brilliant blue dye method, we are fully aware of its drawbacks and will extend their current discussion in section 4.5 by also referring to the shortcomings brought up by previous studies.

Section 2.4: Repetitive from the Hartmann et al. (2020a) paper which in turn was repetitive from Weiler (2001).

That is correct. In both studies the same method was used, since this is a well-established method to analyze brilliant blue dye tracer images and to classify flow types. For the sake of readers who are not familiar with this method we described it here again.
However, we are also open to the possibility of strongly shortening this section to basically citing the used methods and only pointing out specific differences.

Section 2.5: It is completely unclear how disturbed soil samples and soil cores can help reconstruct subsurface flow pathways. Of course, they can give an indication of vertical soil water movement, but not horizontal pathways.

There seems to be a major misunderstanding with respect to the topic of horizontal pathways: as previously stated we only investigated the characteristics of vertical flow paths. We will make sure to clarify this even more in the revised manuscript.
The soil samples were collected for the analyses of soil texture and physical and hydraulic properties since we evaluate the flow path evolution together with soil development and we also think that looking at the soil properties is important when investigating subsurface flow paths.

Section 2.6: Please remember, dye coverage does not equate to flux.

We are not sure how this comment relates to section 2.6 which is about the statistical analysis. However, we are well aware that dye coverage does not quantitatively equate to flux. We will make sure to clearly state this in the revised manuscript.

Section 3.1: I see no connection between these values of vegetation complexity and subsurface flow paths; again, this seems like data looking for a hypothesis link.

As one of our research questions was on the potential usefulness of this geobotanical proxy, that lumps together habitat composition and underground plant traits in explaining subsurface flow paths, we think it is important to provide the results of the vegetation complexity determination. However, we agree that it might be misleading to provide these details here and will move the description of the plot characteristics into the methods section, i.e. we will move and revise the entire section 3.1.

Section 3.2: Much of this section was already covered in the previous two papers by these authors and, once again, other than the connection made between selected soil properties and vegetation complexity, there is no connection to subsurface flow.

In this section we show the results of the soil analyses of the soil samples taken from the choronosequence on the calcareous parent material differentiated by the vegetation complexity. These results got nothing to do with the results from Hartmann et al. (2020a), since they were taken from a completely different chronosequence in a different geology. This first study was conducted on siliceous parent material and it would be dangerous to assume that both soils develop the same way given the strongly different parent material. As soil characteristics can strongly control subsurface flow this soil development has a strong link to the evolution of subsurface flow paths and it therefore seems important to report on the evolution of the soil physical characteristics along the chronosequence. The connection to subsurface flow is addressed in the discussion of the evolution of subsurface flow paths (section 4.2).
Also the data shown in Hartmann et al. (2020b), are not similar. In Hartmann et al. (2020b) the data are published differentiated by depth for the entire moraine and are not interpreted or analysed further, since it is a data publication. The data set presented here is differentiated by vegetation complexity to investigate its possible impact and statistical analyses are carried out both along the age gradient as well as along the complexity gradient as vegetation is known to have an impact on soil weathering.

Section 3.3: These are the most interesting findings of this study and seem to be the most unique findings – i.e., not addressed in the previous two papers. That said, there remain issues of scale associated with the small size of these plots and how representative these are of the broader vegetation complexes and the inferences made herein. Possibly the authors could consider publishing only this part of the paper as a note. The rest of the paper does not strike me as unique. Also, as stated, this part only refers to vertical flow paths and thus has interpretative limits. Please see my comments in several places related to the artificial irrigation applications and the difficulties generalizing these findings as well.

We are glad that the reviewer finds our results of brilliant blue flow pattern interesting. However, we are reluctant to reduce this study to only reporting our observations on the evolution of subsurface flow patterns. We feel that in terms of providing the full picture of flow path evolution we have to accompany these results also with the investigation of soil properties, since these play a crucial role in subsurface water transport. Given the scarceness of these types of detailed studies on flow path evolution as well as the evolution of soil physical characteristics we believe that this second chronosequence study on different parent material is also worthy of publication.

Section 3.4: In an abbreviated version of the paper, which seems appropriate, this section would then be the Discussion. The scale limitations would need to be discussed here.

See our response above why we think it is important to also report the evolution of the soil physical characteristics and thus not abbreviate the manuscript as suggested in the previous comment. We will, however, add a brief discussion on scaling issues to our discussion of the uncertainties in section 4.5.

Section 4.1: This is mostly a rehash of older research, including what has already been presented in these authors papers. Most is not needed.

We find that it is important to put our results into the context of the results of previous studies. This also includes our previous study on siliceous parent material as we cannot simply assume that the soils on different parent materials evolve the same. However, we can easily shorten the comparison with our previous study (which we provided for the sake of completeness) if the editor agrees with this way forward.

Section 4.2: Again, very little new here, compared to what these authors have previously reported. A much more concise version of the material presented from pg. 21, L. 32 to Pg. 22, L. 32 could be included in the Discussion of a modified note or paper, but much of the speculative material and inferences should be removed – e.g., some of the assumption about hydrophobicity, which was not tested.

As stated above we are willing to shorten the comparison with our previous study if the editor agrees that this will improve the manuscript. However, we do think it is important to relate back to these previous results.
Furthermore, our references to possible impacts by hydrophobicity is not just an assumption, since it was actually measured by Maier et al. (2019) and Maier and van Meerveld (2021, WRR in review) and showed and increase with increasing moraine age analog to the increase in preferential flow paths. We will include the reference to these findings in the revised manuscript.

Section 4.3 – Soil structure and texture: Again, this was somewhat covered in the authors prior papers. They mention the inadequacy of the two soil sampling locations relative to the vegetation complexes, and there is still no connection to subsurface flow. I feel this section adds very little.

Again, there seems to be a misunderstanding. While sections 4.1 and 4.2 focus on the evolution of soil characteristics and flow paths with age, sections 4.3 and 4.4 discuss their relationship with vegetation complexity. Neither of the previous papers went into any details in this respect.  In section 4.3, it was not our intention to make a connection to subsurface flow. As we state several times and also make clear within our manuscript, we investigate both the soil development as well as the evolution of the subsurface flow paths. We also stated that it is well known that vegetation has an impact on soil weathering. For that reason, it was our intention to investigate whether different findings in soil properties can be related to vegetation complexity. Of course we had to discuss the mismatch of soil sampling sites to the scale of vegetation complexity estimation and the heterogeneity of vegetation complexity within this scale.

Section 4.3 – Subsurface flow paths: This is mostly a repeat of what was presented earlier in this paper. It does not address 3-D flow, rather vertical pathways. Problems with the small plot size in glaciated terrain should have been anticipated prior to designing this experiment – e.g., the large boulders; pg. 25, L. 12-14). Furthermore, the description of the cause of overland flow that occurred during irrigation (pg. 25, L. 1-10 ) brings into question the artificial irrigation scheme; the explanation provided is does not address this, but rather focuses on the well-known process of surface sealing with no evidence presented. Also, why was this phenomon not observed in the previous study (again, not well explained)?

Please also see our response to the comment above. We only intended to use these dye tracer experiments to investigate vertical subsurface flow paths and never claimed to fully resolve 3D flow. We also argue that our plot size is sufficient for our research goal. It is furthermore also a tradeoff between the desired scale of flow path observation and the required effort for excavation and minimum level of landscape disturbance. However, as stated above we will add a brief discussion on scaling issues to the revised manuscript. Our interdisciplinary field campaign required in both studies a very large number of profile excavations, but only in two cases large sized boulders were found to interfere with the experiment evaluation.

We discuss the impact of artificial irrigation in section 4.5 and will extend this discussion according to our response above.

We also explain why we think, that structural sealing only occurred in this study (calcareous parent material) and not in the previous study (siliceous parent material) by referring to the higher clay content and lower vegetation coverage at the respective moraine at the calcareous parent material. Both crucial points for initiation of structural sealing. In a revised manuscript, we will underline our assumption with the findings of a lower aggregate stability at the moraines of calcareous parent material (Greinwald et al., 2020 in review), which was measured during our field campaign.

Additionally the phenomenon of structural sealing was also observed by Maier and van Meerveld (2021, WRR in review) during larger scale irrigation experiments (briefly mentioned in the method section) on the same moraines in very close proximity to our experimental plots.

Section 4.4: You simply cannot compare the effects of different rain intensities in different biogeoclimatic areas and with different application methods. Thus, this section is of little value.

One of our research questions was focused on the potential impact of irrigation intensity on subsurface flow paths for our study site. We therefore think it is justified and desirable in scientific practice to look at what other studies have found on this topic.

In this section, we also discussed whether our results are directly comparable with those of previous studies. We agree with the reviewer that it is advisable to add 1-2 sentences on the difficulty of comparing the impact of rainfall intensities in different biogeoclimatic areas and will do so in the revised manuscript.

Section 4.5: Pg. 26, L. 9-13, Finally there is some mention of the drawbacks of using Brilliant Blue dye. Also, there is some acknowledgement of the obvious role that high energy water applications had on surface sealing (but why only in this study?) (pg. 26, L. 16-25). This was undoubtedly a major problem in this study design.

As we described in line 16 page 26, due to weather conditions the sprinkler had to be guided close to the soil surface at the two youngest moraines, which was not the case in the previous study. As already described in the manuscript and here in the response above: due to lower clay contents and a higher vegetation coverage we did not encounter this problem during the first study conducted on siliceous parent material, which again underlines our statement that parent material matters and a follow on study on a different geology is worth showing.

Section 5: All previous comments apply; in addition, while the statement on pg. 26, L. 30 may be true ("This shows that the influence of preferential flow paths increases with soil age"), the lack of robust 3-D evidence and a larger-scale perspective put this in question. I did not think the authors made a good connection (at appropriate scales) between vegetation complexity and subsurface flow paths, but now this is stated in the Conclusions as "We saw a direct relationship between vegetation complexity and subsurface flow paths at the old moraines and a relationship of vegetation complexity and soil properties at the 110, 4 900, and 13 500-year-old moraines" – this simply was not verified. Certainly, some inference could be made for vertical pathways, but even these were rather subjective. One of the concluding sentences – "…we still suggest that a more in-depth consideration of vegetation characteristics beyond coverage and land use types will provide useful insights for hydrological process research", leaves the reader hanging and asking what was really accomplished here that was not reported in the previous two papers by these authors. And I do not see convincing evidence of the stated feedback cycle of the hydro-pedogeomorphological system.

While we do agree that we should rephrase the first sentence cited above to only refer to "vertical preferential flow paths", we do not agree with the overall gist of this comment. In this follow on study we conducted a study on a glacier forefield of calcareous parent material. We see from our results but also from previous studies by different authors that parent material matters in soil development and flow path evolution and that it is worth repeating similar studies on different geologies to get a better understanding of the feedback cycle of the hydro-pedogeomorphological system.
With our work in both glacier forefields, we provide important but currently still rare data and observations which will help to ensure proper handling of (subsurface) hydrologic processes and their role within the feedback cycle of the hydro-pedo-geomorphological system when it comes to soil and landscape evolution modeling.
The repetition of our experiments in a different environment is part of a scientifically recognized practice, to avoid undue generalization of the knowledge gained at one specific site and to identify possible influencing factors. We are thus convinced that it is worth providing these additional insights and observations to the community (similarly as hydrological catchment studies have been carried out in various catchments with different characteristics to investigate different controls on runoff generation).

**References**

Amundson, R., Richter, D. D., Humphreys, G. S., Jobbágy, E. G., and Gaillardet, J.: Coupling between biota and earth materials in the critical zone, Elements, pp. 327–332, 2007.

Bachmair, S., Weiler, M., and Nützmann, G.: Controls of land use and soil structure on water movement: Lessons for pollutant transfer through the unsaturated zone, Journal of Hydrology, 369, 241–252, doi.org/10.1016/j.jhydrol.2009.02.031, 2009.

Greinwald, K., Gebauer, T., Treuter, L., Kolodziej, V., Musso, A., Maier, F, Lustenberger, F., Scherer-Lorenzen, M.: Vegetation dynamics drive the evolution of soil aggregate stability during the first 14,000 years of soil development in the Swiss alps. 2021, in review

Kan, X., Cheng, J., Hou, F.: Response of Preferential Soil Flow to Different Infiltration Rates and Vegetation Types in the Karst Region of Southwest China, Water,12,1778, doi.org/10.3390/w12061778, 2020.

Keith, A., Henrys, P., Rowe, R., and Mcnamara, N.: Technical note: A bootstrapped LOESS regression approach for comparing soil depth profiles, Biogeosciences, 13, 3863–3868, doi.org/10.5194/bg-13-3863-2016, 2016.

Ivory, S. J., McGlue, M. M., Ellis, G. S., Lézine, A.-M., Cohen, A. S., and Vincens, A.: Vegetation Controls on Weathering Intensity during the Last Deglacial Transition in Southeast Africa, PLOS ONE, 9, e112 855, doi.org/10.1371/journal.pone.0112855, 2014.

Maier, F., van Meerveld, I., Greinwald, K., Gebauer, T., Lustenberger, F., Hartmann, A., and Musso, A.: Effects of soil and vegetation development on surface hydrological properties of moraines in the Swiss Alps, CATENA, 187, 104 353, doi.org/10.1016/j.catena.2019.104353, 2020.

Maier, F., and van Meerveld, I.: Long-term changes in runoff generation mechanisms for two proglacial areas in the Swiss Alps I: Overland Flow, Water Resources Research, 2021, in review

Stumpp, C. and Maloszewski, P.: Quantification of preferential flow and flow heterogeneities in an unsaturated soil planted with different crops using the environmental isotope $\delta^{18}O$. Journal of Hydrology, 394, 407–415, doi:10.1016/j.jhydrol.2010.09.014, 2010.

---

## Author Comment (AC2)

**Response to Reviewer comments**

**Response to Reviewer 2**

**General Comments**
Hartmann et al. present a generally interesting study on infiltration capacities across a moraine chronosequence, where each chronosequence is divided into three levels of vegetation cover complexity and receives three different water application intensities, resulting in 36 different water applications. However, the experiment appears to be (more or less) a replica of a previous study (Hartmann et al. 2020a&b), with the main difference apparently being the parent material, which is calcareous in this manuscript and siliceous in the previous ones, and an apparent focus on vegetation and rainfall intensity influences.
Even some of the figures are largely identical. It is not entirely clear to me what the new contribution of this manuscript is over the previously published study.
Additionally, I do have some concerns with the general study layout and possible interpretations. Each plot is divided into three 50cm wide zones with different rainfall application intensities. These zones are not physically separated from each other and to prevent interaction during application in one zone, the remaining two are covered. This still leaves room for interaction near the zone boundaries where water can be drawn laterally into the drier soil of a neighboring zone. The authors acknowledge as well (see below) that overland flow on some plots might have infiltrated near the zone boundaries, leading to increased infiltration there. Then there is the question of the ages in the chronosequence. The two younger sites are 110 and 160 years old. 50 years difference is not much in a soil chronosequence, especially considering that the other soils are 4900 und 13000 Jahre alt. Unless I missed it, I did not see an explanation of what the authors expect in those 50 years to have happened to the soil.
If this were the authors' sole publication on the topic, I would probably just consider major revisions (i.e., shortening and some restructuring). Given the other two publications, I am having difficulty seeing the novelty in this manuscript, though, and am unfortunately leaning toward rejection.

**Response to General Comments**
The authors would like to thank the reviewer for spending her/his time on this review.

We respect but regret the decision of the reviewer not to support a publication of our manuscript in HESS.

As we do not agree with this assessment and are convinced that this is largely based on a misunderstanding we provide detailed and hopefully convincing responses to the general and specific comments below.

First of all, we would like to emphasize once again that our study is deliberately a follow-up study. In our first study on the co-evolution of flow paths and soil properties along a chronosequence of hillslopes in a glacial forefield, the experiments were carried out on siliceous parent material. To our knowledge this was the first systematic study on subsurface hydrologic flow path evolution during the first 10000 years of landscape evolution.
The role of hydrologic processes, especially subsurface preferential flow, is mainly missing in soil and landscape evolution modeling, which is mostly due to the lack of observations on temporal changes and dynamics of subsurface flow paths (van der Meij et al., 2018).

With our first study we provided rare data and observations on this topic. However, it is also a very important scientific practice to replicate such a study in different geologies for the verification and generalization of the gained knowledge. Simply taking this single study and assuming that the same

processes/evolution occur in a totally different geology seems risky and unscientific. In our opinion, this applies above all to the complex interplay within the hydro-pedo-geomorphological system, where the parent material has a significant influence.  Due to our follow-up study it became clear, that parent material does indeed matter in soil development and flow path evolution, since we found significant differences in flow paths after 10000 years of landscape evolution. This is an important finding for the investigation of the feedback cycle of the hydro-pedogeomorphological system. We will improve the highlighting of these findings in the revised manuscript, since it was obviously not clear enough.

To facilitate comparisons between the two studies, we decided to keep the design of the graphics from the first study for reasons of consistency. This should make it easier for the reader to compare the data and results from both publications with one another. The basic design of some graphics is therefore identical in both publications; the data presented therein are not. The contribution of this study lies clearly in the completion of the rare observations when it comes to the development of subsurface hydrology within landscape development. Our two studies provide these rare data and observations, which will help to ensure proper handling of (subsurface) hydrologic processes and their role within the feedback cycle of the hydro-pedo-geomorphological system when it comes to soil and landscape evolution modeling.

We would also like to make it clear again that the experiments in both studies are quite similar, but differ in the following points: In addition to the development of the preferential flow paths with moraines age (investigated in both studies), the development of flow paths depending on irrigation amount was the second focus in the first study, while the current study focused on the irrigation intensity. Furthermore, within this current study we also set the focus on the influence of vegetation complexity. This was only touched on very superficially in the previous study.

Reasons why we claim that the here submitted study is a valuable and novel contribution to scientific knowledge (despite the previous study in a different geology (study 1) and the previous publication of the raw soil physical data in a data publication):

a) The focus of the two studies differs: age sequence and irrigation *amount* in study 1, age sequence, irrigation *intensity* and vegetation complexity in study 2.
b) The parent material of the previous study is siliceous, the parent material here is calcareous. Assuming that all geologies result in similar processes and developments is dangerous and we show here that this assumption is wrong. Given that study 1 was praised as novel from a pedological perspective and interesting to the community and the data set is not only unique, but also essential for any quantitative modelling of water and element balances of such soil ecosystems and highly relevant also for neighboring disciplines we are convinced that a follow-up study in a different geology and with a different focus is still innovative and worth publishing.
c) The smaller age difference between the two youngest moraines in study 2 furthermore shows that significant development and changes can already occur over a period of only 50 years.
d) The soil physical data of the site was published as a data publication. However, data publications do not contain any interpretation and usually no statistical analysis. In the here submitted study 2, we present statistical tests showing which differences between classes and variables are statistically significant and which are not. We also show different groupings compared to the overview plots of the data publication.
e) The similar design of some of the figures in study 1 and 2 is supposed to facilitate the comparison of the two studies and should not be mistaken as a lack of novelty.
f) Additional methodological novelty: Study 2 furthermore contains a novel comparison of the dye profiles based on the bootstrapped LOESS regression (BLR). This approach to compare two data

sets of profile observations was first developed by Keith et al. (2016) who used it to compare profiles of soil organic carbon. The method has never been applied to dye profiles of irrigation experiments.

With regard to the study design, as correctly described, we occasionally suspected that at plots with surfaces influenced by structural sealing (as a result of the high intensity of sprinkling), surface runoff from the irrigated plot onto neighboring, non-irrigated areas occurred. This occurred to a small extent in 5 out of 36 irrigation experiments. In general, after each irrigation, a possible runoff to neighboring areas was checked, but this was only observed at the plots with very sparse vegetation (i.e. the youngest plots). Thus, we can rule out a general influence. We will clarify this in the revised manuscript on page 26, specifically providing the information that only 5 out of 36 experiments were affected.

Our study is an interdisciplinary study in which we work together with other disciplines. Therefore, the selection of the age classes was not only based on hydrological aspects, but also under aspects of geobotany and geomorphology.
Regarding the age gap between the two youngest moraines in this study, we want to point out that we found large changes between the 30 and the 160 year old moraines in our previous study, as developments are especially fast during these early stages. Even between the two youngest moraines in the here presented study which only have an age difference of 50 years, significant changes in flow responses could be observed (Figures 8, 9, A1). We will highlight these short-term changes more clearly in the revised manuscript as this is indeed another piece of important additional information: Changes over a short time span of only 50 years are significant. However, the original reason of selecting the 110 year old as youngest moraine was mainly the result of the local conditions at this site. The actual goal was to select age groups that were as identical as possible in both chronosequences (i.e. the two geologies). This was not entirely possible for the youngest moraines at both locations (with ages of 30 years at siliceous parent material and 110 years at calcareous parent material). The choice of the 110 year old moraine as the youngest moraine is the result of the local conditions at this site, as no adequate moraine with an age of 30 years could be identified that also ensured comparability in terms of elevation and microclimate. We therefore had to compromise and selected the moraine with an age of 110 years as our youngest moraine at this site (Musso et al., 2019).

The reviewer suggests shortening and restructuring of the manuscript. We will thoroughly revise the manuscript with this in mind, while at the same time providing all relevant information.

Moraines are a special type of cover and pedogenesis. Can you hypothesize what can be expected in soils that formed not from direct glacial processes?

The results of our two studies have shown that the observations cannot be simply generalized to other locations. The combination of our two studies has shown that the parent material has a significant influence on the dynamics of subsurface flow paths. If we consider soils that do not form on glacial till, it can also be assumed that other time scales must be considered, since weathering may not progress as quickly as with loose glacial till. In addition, other influencing factors such as vegetation and climate must be taken into account. We assume that at least in the upper centimeters, where the influence of weathering is strongest and also leads to a strong reduction in the grain size of other raw materials, heterogeneous flow patterns with a high proportion of finger flow might are dominant after a few years, which might transits to macropore flow at highly developed soils. We will include a sentence in this respect in the revised manuscript.

Fig 5: Are the figures the mean of the five excavated profiles?

Yes, it is the mean volume density as described on page 7, line 16. We will also include this information in the figure caption in the revised manuscript.

P21 L10-11: Is this purely based on the parent material or maybe also a function of landscape position, e.g., aspect, slope, etc.?

The difference is largely due to the different parent material and the associated soil chemistry. Other environmental factors were kept as similar as possible. We will clarify this in the revised manuscript.

P22 L26-28: For the sake of comparability, would the "finger flow and macropore flow (high interaction)" class be classified as macropore flow in the previous study? (only based on patterns, even in the absence of actual macropores)

This might have been a misunderstanding: We used the same classification scheme in both studies, which makes the flow type distributions directly comparable. In both studies the major component of this joint flow type class was finger flow.

A few more details on this flow type class: We introduced this joint flow type class in our previous study as mentioned on page 22 line 27 and explained on page 7 line 31-33, since in our previous study we observed narrow finger like flow paths that were misclassified as macropore flow with high interactions when using the original classification scheme. Macropores with high interactions could not be verified based on the photographs. In order not to completely rule out their occurrence in alpine soils, we decided to put both flow types in a joint class and to identify the major component based on the observations.

P22 L30-32: Given that the top layers contain hydrophobic material in both studies and no overland flow is observed, wouldn't that suggest that most water should make it through this hydrophobic layer?

That is correct. Since little or no overland flow was observed, the water must have infiltrated vertically through the hydrophobic layer. We did not assume that the hydrophobicity of the material generally prevents infiltration, but is also heterogeneous and together with the microtopography of the surface

leads to heterogeneous infiltration patterns and thus to the increased development of preferential flow paths in the form of finger flow. We will clarify this in the revised manuscript.

 This raises a question about the experiment setup. If I understand correctly, the zones for different application intensities were neither separated by a non-irrigated space in between, not by some barrier installed into the soil profile that could have prevented surface flow onto the adjacent zone? If this is the case, couldn't it be possible that the areas where the transition from one zone to the next happens simply receive more water than the rest of the zones? If deeper infiltration is observed below the transitions, that could be a result of more water infiltrating and thus being able to reach greater depths.

During the irrigation experiment we covered the non-irrigated neighboring subplots with a tarpaulin that physically protects the surface from the irrigation water. At the youngest moraines, where the direct application of water by spraying might have caused structural sealing on the currently irrigated (bare) plot, some of this water moved laterally along the surface (and under the tarpaulin) onto the neighboring plot which was still covered and thus not affected by spraying and the resulting structural sealing. So in these locations there might have been slightly more water input, but also, we here see the infiltration under less disturbed circumstances. However, the amount of water running off to the sides is not substantial due to the low inclination of the plots to the sides. We will include this information in the revised manuscript.
Nevertheless, we checked for this possible lateral runoff to neighboring plots after each irrigation, but only observed this to a small extent at bare plots at the young moraines. This was an issue observed in 5 out of 36 experiments. However, the deep percolation at the transition zones thus show that rapid vertical deep infiltration is possible at this age class, when the soil surface is unaffected by sealing as a result of the spraying action.

Keith, A., Henrys, P., Rowe, R., and Mcnamara, N.: Technical note: A bootstrapped LOESS regression approach for comparing soil depth profiles, Biogeosciences, 13, 3863–3868, https://doi.org/10.5194/bg-13-3863-2016, 2016.

Musso, A., Lamorski, K., Sławiński, C., Geitner, C., Hunt, A., Greinwald, K., and Egli, M., 2019: Evolution of soil pores and their characteristics in a siliceous and calcareous proglacial area, CATENA, 182, 104 154, https://doi.org/10.1016/j.catena.2019.104154.

van der Meij, W., Temme, A., Lin, H., Gerke, H., and Sommer, M., 2018: On the role of hydrologic processes in soil and landscape evolution modeling: concepts, complications and partial solutions, Earth-Science Reviews, 185, 1088 – 1106, https://doi.org/https://doi.org/10.1016/j.earscirev.2018.09.001.